# Ensemble reconstruction of missing satellite data using a denoising diffusion model: application to chlorophyll $a$ concentration in the Black Sea

Alexander Barth[1], Julien Brajard[2], Aida Alvera-Azcárate[1], Bayoumy Mohamed[1], Charles Troupin[1], and Jean-Marie Beckers[1]

[1]GHER, FOCUS, University of Liège, Liège, Belgium
[2]Nansen Environmental and Remote Sensing Center and Bjerknes Centre for Climate Research, Bergen N5007, Norway

**Correspondence:** A. Barth (a.barth@uliege.be)

**Abstract.** Satellite observations provide a global or near-global coverage of the World Ocean. They are however affected by clouds (among others), which severely reduce their spatial coverage. Different methods have been proposed in the literature to reconstruct missing data in satellite observations. For many applications of satellite observations, it has been increasingly important to accurately reflect the underlying uncertainty of the reconstructed observations. In this paper, we investigate the use of a denoising diffusion model to reconstruct missing observations. Such methods can naturally provide an ensemble of reconstructions where each member is spatially coherent with the scales of variability and with the available data. Rather than providing a single reconstruction, an ensemble of possible reconstructions can be computed, and the ensemble spread reflects the underlying uncertainty. We show how this method can be trained from a collection of satellite data without requiring a prior interpolation of missing data and without resorting to data from a numerical model. The reconstruction method is tested with chlorophyll $a$ concentration from the Ocean and Land Color Instrument (OLCI) sensor (onboard the satellites Sentinel-3A and Sentinel-3B) on a small area of the Black Sea and compared with the neural network DINCAE (Data-Interpolating Convolutional Auto-Encoder). The spatial scales of the reconstructed data are assessed via a variogram, and the accuracy and statistical validity of the produced ensemble reconstructed are quantified using the continuous ranked probability score and its decomposition into reliability, resolution and uncertainty.

## 1 Introduction

At any given time, about 75% of the ocean surface is covered by clouds (Wylie et al., 2005) which are opaque to electromagnetic radiation in the visible and infrared spectrum. Many satellite sensors rely on this part of the spectrum to measure, for example, sea surface temperature and ocean color. Besides clouds, other reasons for missing data include atmospheric dust, sun glint contamination, limited swath width, and high sensor-zenith angle (Feng and Hu, 2016; Mikelsons and Wang, 2019; Alvera-Azcárate et al., 2021). The amount of missing data in satellite observations can therefore be substantial.

Several methods have been proposed in the past to reconstruct missing data in satellite images, such as EOF-based (Empirical Orthogonal Functions) methods like Data Interpolating Empirical Orthogonal Functions (DINEOF, Alvera-Azcárate et al., 2016; Alvera-Azcárate et al., 2021; Pujol et al., 2022), optimal interpolation (*e.g.* Reynolds et al., 2007), and Kriging (*e.g.* Saulquin et al., 2011). More recently, neural network-based techniques, such as the Data-Interpolating Convolutional Auto-Encoder (DINCAE, Barth et al., 2020; Han et al., 2020; Ji et al., 2021; Jung et al., 2022; Barth et al., 2022; Luo et al., 2022) and other neural networks with a U-Net architecture (Ronneberger et al., 2015) like those described by Hong et al. (2023) as well as marked auto-encoders (Goh et al., 2023), have been applied to this problem. The input of these neural networks is typically a satellite image with missing data and the output is the reconstructed full field. Then the neural network is trained by being fed pairs of images (with and without clouds, or with some clouds and with even more clouds) so that the neural network learns the mapping between an image affected by clouds and a clear image.

For satellite images where all missing data have been reconstructed, it is clear that the error of the reconstructed and initial missing pixels is typically larger than the error of the original pixels. In optimal interpolation and Kriging, this error is represented by the *a posteriori* error covariance. However, these methods assume that the errors can be described by a Gaussian distribution and that the underlying error covariances of the observations and the first guess are perfectly known. In practice, the error covariance of the first guess (the *a priori* error covariance) is often described as an isotropic function depending only on the distance between two points. In addition, these methods assume that the observations and the first guess are unbiased and independent.

For DINCAE (Barth et al., 2020, 2022), the estimation of the error variance is part of the training process and does not require precise knowledge of the error statistics of the input data. For every pixel, an estimate of the reconstructed value and its error variance is provided. During the training process, the likelihood of the actual measurement is maximized by assuming that the error is Gaussian distributed. This gives a pointwise estimate of the error variance and validation with independent data shows that the expected error variance is reliable. However, this approach does not give us any information about how the error is correlated in space (and time). This additional information is crucial for computing the expected error of derived quantities that combine satellite data from different spatial locations. For example, this is the case when computing an average quantity over a given area.

Another issue, when the model is forced to provide a single reconstruction, is that the results are often too smooth, as small scales under clouds are of course not resolved when the cloud coverage has a given spatial extension (and only large scales can be estimated using available data). Since multiple images would be consistent with the partial information present, a neural network trained to minimize e.g. the mean square error, would then implicitly produce the average of all these possible states. For example, if the exact position of a front is not visible in a satellite image, a reconstructed image would have the tendency to smooth out the front as it is implicitly the average of multiple images with the front in different positions. Consequently, this means that small scale information cannot be adequately retained.

Therefore, instead of creating a single reconstruction for each pixel (with the associated error variance), it would be preferable to produce an ensemble of likely reconstructions (based on the available data), as is the case with ensemble modeling and the Ensemble Kalman Filter (Evensen, 2009). The expected error of a derived quantity (for example total amount of surface chlorophyll in a given area) is then given directly by the ensemble statistics where this derived quantity is computed for each member of the ensemble individually.

The denoising diffusion models (*e.g.* Ho et al., 2020) belong to the family of generative algorithms like Generative Adversarial Networks (Goodfellow et al., 2016). Contrarily to deterministic neural networks, in which the primary objective is to learn a mapping function between input features and a desired output, generative models aim to produce samples from the same distribution as the training data. In general, such probability distribution cannot be expressed explicitly in closed form. In many studies (*e.g.* Dhariwal and Nichol, 2021; Bayat, 2023), it has been shown empirically that diffusion models tend to have better diversity than Generative Adversarial Networks which is an important property if one wants to represent the uncertainty of the reconstruction of incomplete satellite data.

Often diffusion models use additional information (for example text description or an image) in order to guide the generation process during the reverse diffusion process (i.e. the image generation process). This guidance can be implemented in different ways. One can either use a classifier to steer the generation process (Dhariwal and Nichol, 2021). A classifier is a neural network which associates a label (typically a text description) to an image. However, it is important that the pre-trained classifier is suitable for noisy images as generated during the reverse diffusion process.

In the classifier-free guidance algorithm (Ho and Salimans, 2022), the neural network denoising the images also depends explicitly on the class label. While training the neural network, this class label is sometimes replaced by a null label (i.e. a vector with all elements equal to zero). As a result the trained neural network can either denoise *any* image of the training dataset (when given the null label) or a specific subset of the training dataset (matching the provided label). During sampling the reverse diffusion is steered by a scaled difference between the noise predicted knowing the label and the noise predicted with a null label and therefore enhancing the similarity of the generated image with the provided label.

Denoising diffusion models have also been used for increasing the resolution (Saharia et al., 2021) and for in-painting. Lugmayr et al. (2022) apply the forward and reverse diffusion process iteratively to fill in the missing region. However, for these approaches the diffusion model must be trained on a large collection of complete images.

In section 2, we will introduce the denoising diffusion framework which is the basis of this work. The data will be presented in section 3. The denoising diffusion framework will be adapted in section 4 to handle missing data during the training and to produce reconstructed images based on partial data. The results will be discussed and validated in the section 5 and compared to the DINCAE method. Conclusions will be presented in section 6.

## 2 Denoising diffusion model

The denoising diffusion models (Ho et al., 2020) use a quite different approach than traditional applications for neural net-
works, as their goal is to generate an image that comes from the same (but not explicitly known) distribution as the training
data. This inherently stochastic generation process gives us an appropriate framework to provide an ensemble of possible states.

The present description closely follows Ho et al. (2020). The general idea is that we start with a clear image $\mathbf{x}_0$ (later we
will discuss the case where all training images contain clouds) and then progressively add noise. Without loss of generality, we
assume that $\mathbf{x}_0$ is scaled such that every element is of the order of 1. In practice, we remove the mean and divide the anomalies
by the standard deviation. The mean and standard deviation are here single scalars computed over the whole training dataset.
We did not compute a different mean and standard for every image.

The diffusion process is a Markov process as every image $\mathbf{x}_t$ (considered here as a flat vector) depends only on the previous
image $\mathbf{x}_{t-1}$ in this chain. We degrade the image $\mathbf{x}_{t-1}$ by adding noise $\mathbf{z}_t$ ($\mathbf{z}_t \sim \mathcal{N}(0,\mathbf{I})$) scaled by the parameter $\beta_t$ (with
$0 < \beta_t < 1$). The variance of added noise ($\beta_t$) typically increases at the diffusion step $t$ increases. Note that the diffusion step
is completely unrelated to the acquisition time of the satellite data. At the same time, we scale the image $\mathbf{x}_{t-1}$ so that the
combination $\mathbf{x}_t$ remains of unit variance:

$$\mathbf{x}_t = \sqrt{1-\beta_t}\mathbf{x}_{t-1} + \sqrt{\beta_t}\mathbf{z}_{t-1} \tag{1}$$

The level of degradation in the image $\mathbf{x}_t$ increases as the diffusion step $t$ increases. This Markov process has the following
transition probability (also called forward diffusion kernel):

$$q(\mathbf{x}_t|\mathbf{x}_{t-1}) = \mathcal{N}\left(\mathbf{x}_t; \sqrt{1-\beta_t}\mathbf{x}_{t-1}, \beta_t\mathbf{I}\right) \tag{2}$$

The linear combination of two Gaussian distributed variables is also Gaussian distributed. Therefore, we can compute the
transition probability $q(\mathbf{x}_t|\mathbf{x}_0)$ in closed form (Ho et al., 2020):

$$q(\mathbf{x}_t|\mathbf{x}_0) = \mathcal{N}\left(\mathbf{x}_t; \sqrt{\bar{\alpha}_t}\mathbf{x}_0, (1-\bar{\alpha}_t)\mathbf{I}\right) \tag{3}$$

where $\bar{\alpha}_t = \prod_{s=1}^t \alpha_s$ and $\alpha_t = 1 - \beta_t$. The parameters $\bar{\alpha}_t$ and $\alpha_t$ generally depend on the diffusion step $t$. All elements $\alpha_t$
are smaller than 1, therefore, $\bar{\alpha}_t$ tends to zero as $t$ increases. The image $\mathbf{x}_t$ will become more and more similar to an image
with Gaussian noise as $t$ increases. The last image $\mathbf{x}_T$ approximately follows a Gaussian distribution with zero mean and an
identity matrix as covariance:

$$q(\mathbf{x}_T) \approx \mathcal{N}(\mathbf{x}_T; 0, \mathbf{I}) \tag{4}$$

## 2.1 Reverse process

If the forward transition kernel is a Gaussian distribution, the distribution of the reverse transition kernel is also a Gaussian distribution in the limit of small steps sizes $\beta_t$, i.e. in the limit where the discrete diffusion process tends to the continuous diffusion (Feller, 1949; Sohl-Dickstein et al., 2015). The Markov chain for the reverse process begins with a Gaussian Distribution random variable with zero mean and unit variance:

$$p(\mathbf{x}_T) = \mathcal{N}(\mathbf{x}_T; 0, \mathbf{I}) \tag{5}$$

The reverse process is also a Markov process involving the transition probabilities $p_{\boldsymbol{\theta}}(\mathbf{x}_{T-1}, \mathbf{x}_T)$ and a certain number of model parameters $\boldsymbol{\theta}$ to be determined:

$$p_{\boldsymbol{\theta}}(\mathbf{x}_{T-1}) = \int p_{\boldsymbol{\theta}}(\mathbf{x}_{T-1}, \mathbf{x}_T) d\mathbf{x}_T \tag{6}$$

Formally, the probability of the clear image $\mathbf{x}_0$ is obtained by combining the probability of all possible trajectories $\mathbf{x}_{0:T}$ leading to the image $\mathbf{x}_0$:

$$p_{\boldsymbol{\theta}}(\mathbf{x}_0) = \int p_{\boldsymbol{\theta}}(\mathbf{x}_{0:T}) d\mathbf{x}_{1:T} \tag{7}$$

The parameters $\boldsymbol{\theta}$ will be determined by maximizing the expected probability $p_{\boldsymbol{\theta}}(\mathbf{x}_0)$, or equivalently by minimizing the negative logarithm of this probability:

$$L = E\left[-\log(p_{\boldsymbol{\theta}}(\mathbf{x}_0))\right] \tag{8}$$

In practice, the integral is intractable as it would require an integration over a very high-dimensional space. It can be shown that $L$ is always smaller than the so-called evidence lower bound $L_{\text{elb}}$ (Sohl-Dickstein et al., 2015) using Jensen's inequality (Jensen, 1906) and the Bayes' theorem:

$$E\left[-\log(p_{\boldsymbol{\theta}}(\mathbf{x}_0))\right] \leq -E\left[\log\frac{p_{\boldsymbol{\theta}}(\mathbf{x}_{0:T})}{q(\mathbf{x}_{1:T}|\mathbf{x}_0)}\right] = -E\left[\log\left(p(\mathbf{x}_T)\prod_{t=1}^{T}\frac{p_{\boldsymbol{\theta}}(\mathbf{x}_{t-1}|\mathbf{x}_t)}{q(\mathbf{x}_t|\mathbf{x}_{t-1})}\right)\right] = L_{\text{elb}} \tag{9}$$

where the latent variables (i.e. unobserved variables) are here the whole trajectory except the first state ($\mathbf{x}_{1:T}$). Rather than minimizing $L$, the quantity $L_{\text{elb}}$ is minimized instead. Ho et al. (2020) showed that this leads, after some simplifications, to the following cost function for training the neural network ($\epsilon_{\boldsymbol{\theta}}(\mathbf{x}, t)$), for any step $t$ and for any sample $\mathbf{x}_0$ from the training dataset:

$$J(\boldsymbol{\theta}) = ||\boldsymbol{\epsilon} - \boldsymbol{\epsilon_\theta}(\sqrt{\bar{\alpha}_t}\mathbf{x}_0 + \sqrt{1 - \bar{\alpha}_t}\,\boldsymbol{\epsilon}, t))||^2 \tag{10}$$

where $\boldsymbol{\epsilon}$ is the accumulated noise added during the forward process. The weights $\boldsymbol{\theta}$ of the neural network are updated using the gradient of the previous loss function. A trained neural network can then be used to create other samples $\mathbf{x}_0$ by solving the following equation backwards where the initial image $\mathbf{x}_T$ and $\mathbf{z}$ follows a normal distribution:

$$\mathbf{x}_{t-1} \;\; = \;\; \frac{1}{\sqrt{\alpha_t}}\left(\mathbf{x}_t - \frac{1 - \alpha_t}{\sqrt{1 - \bar{\alpha}_t}}\boldsymbol{\epsilon_\theta}(\mathbf{x}_t, t)\right) + \sigma_t \mathbf{z} \tag{11}$$

where the noise term $\sigma_t$ is equal to $\sqrt{\beta_t}$. This algorithm will be extended in Section 4.1 to handle clouded images.

## 3 Data

To illustrate the application of the denoising diffusion model, we use the daily L3 satellite chlorophyll $a$ concentration of the Black Sea at a spatial resolution of 300 m from the Copernicus Marine Service (Zibordi et al., 2015; Kajiyama et al., 2019; Lee et al., 2002; European Union-Copernicus Marine Service, 2022) using data from the Ocean and Land Color Instrument (OLCI) sensor onboard the Sentinel-3A and Sentinel-3B satellites. On average, the amount of valid data over the ocean is 30% and shows a clear seasonal cycle (Figure 1). The marked increase of data after 2019 is due to the availability of Sentinel-3B data. We use data from April 26, 2016 to August 31, 2023 of this chlorophyll $a$ concentration dataset. For the training data, we use data up to the date August 31, 2021.

The aim of the study is to test different methods on a problem with relatively small images which allow us to test many different hyperparameters. The training data is therefore split horizontally in tiles with 64 x 64 grid cells. Only tiles with at least 20% valid data (i.e. non-clouded pixels) are used for training to reduce training time. In total, there are 851926 images (after splitting the data into tiles) for training.

The validation dataset is composed of the 12 months of data between September 1, 2021 to August 31, 2022. The following 12 months (from September 1, 2022 to August 31, 2023) are used as test data. We only consider the region 28.56979°E - 28.80623°E and 43.64238°N - 43.81242°N (corresponding to a $64 \times 64$ grid at 300 m resolution) for validating and testing the neural network (while the data from the whole Black Sea is used for training), as the other considered method (DINCAE) has only been tested so far with a fixed location. This is a relatively small area, but it allowed us to perform several tests with different network configurations (Figure 2).

A coastal area was chosen because the dynamics there are more complex than in offshore waters. For the validation and test data, we randomly took the cloud mask from other time instances to mask additional grid cells which will be used for validation. Only images with a cloud mask between 15% and 35% of the missing data were considered as an additional mask to obtain a sufficient number of "clouded" pixels and to reduce the risk that an image is masked almost entirely. We verified that, neither in the validation nor in the test dataset, were images masked entirely after applying the cloud mask.

All the data is log transformed (base 10) and the units are to be understood as $\log_{10} \text{mg m}^{-3}$.

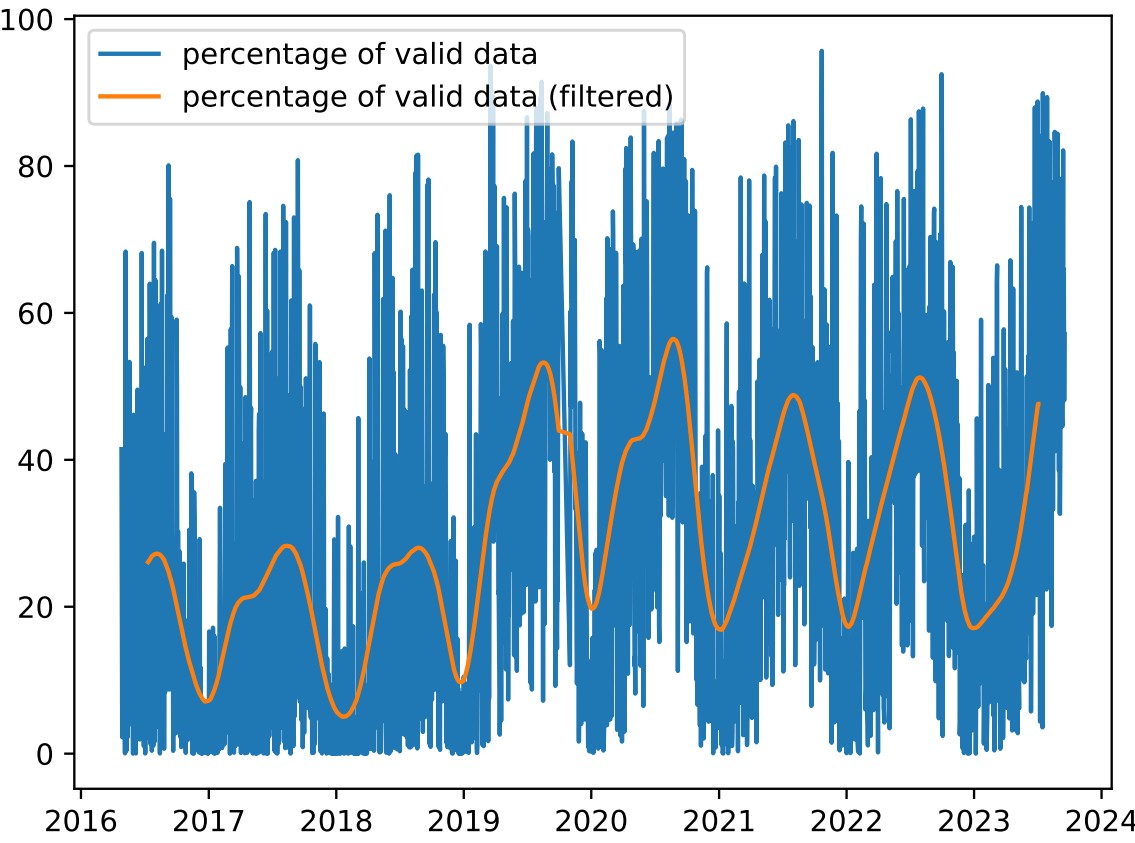

**Figure 1.** Percentage of valid data over time in each satellite image for the Black Sea dataset and percentage of valid data filtered by a Gaussian filter (with a standard deviation of 30 days).

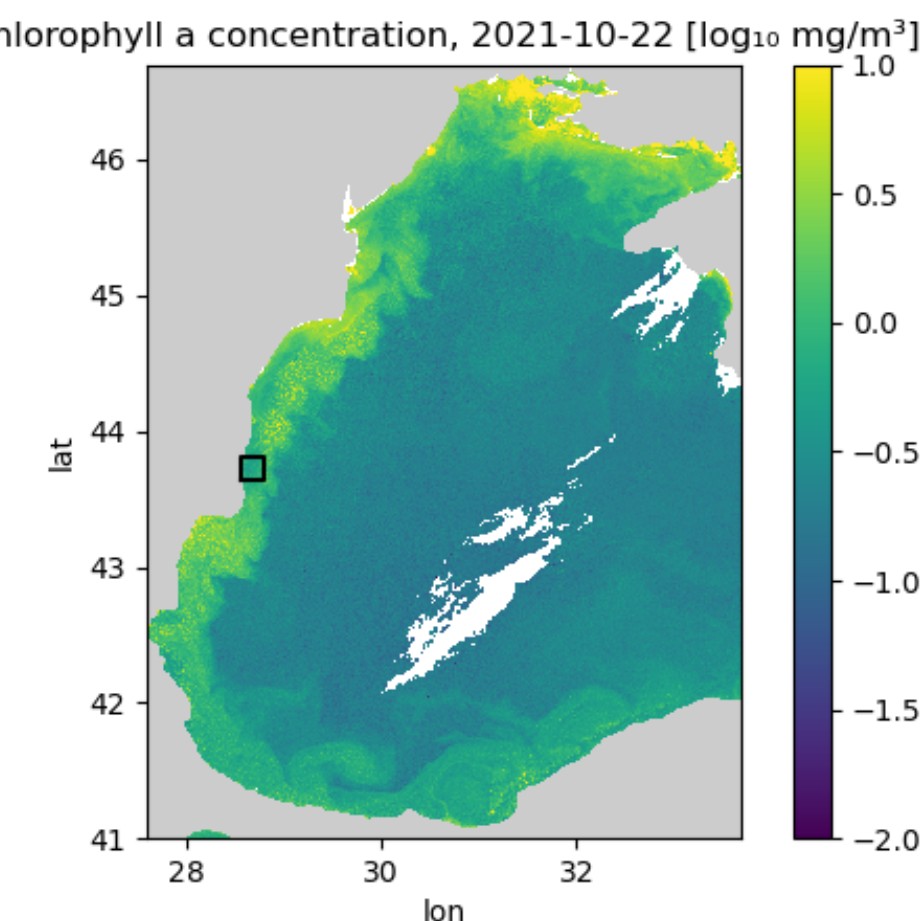

**Figure 2.** Western part of the Black Sea. The black square corresponds to the area where the methods are validated and tested (units: $\log_{10}$ mg m$^{-3}$).

 ## 4  Method

### 4.1  Training with clouded images

The training approach by Ho et al. (2020) assumes that we have a large training dataset with clear images. Unfortunately, for satellite observations, the clouds are so common that it would be difficult to create such a dataset. If the data was previously interpolated, then there is the risk that the neural network would also learn potential interpolation artifacts. Alternatively, the neural network could also be trained with data from a numerical model. But even in this case, the neural network would also learn biases and errors present in the model. When validating models with satellite observations, it is generally preferable that

the satellite observation is independent of a numerical model. Therefore, we are aiming to extend the approach of Ho et al. (2020) to train using images including clouds.

It is important to note that all operations in the training and sampling algorithms (equations 1, 10 and 11) are only pointwise operations (i.e., operations that apply to each grid cell independently) that do not involve the neighboring grid cells, except for the neural network which ensures spatial coherence. The spatial coherence is mainly due to the convolutional layers whose weights have been trained to provide the same spatial structure as in the training dataset. Rather than working with a global step $t$ valid for a whole image, we consider the case where every pixel can be in a different state of degradation. The noise

schedule $\beta_t$ is a freely selectable list of parameters. For the following approach, we impose that $\beta_0 = 0$, which means that the noise is effectively added only at step 1 and later but not at step zero.

For a training image that contains clouds, we consider clouded pixels initially at the fully degraded level $t = T$ (i.e., normally distributed random noise) and clear pixels at the non-degraded level $t = 0$ (i.e., pixels as measured by the satellite). During training, for each image of the training dataset, a different image is randomly selected (also from the training dataset) and its

cloud mask is used to degrade clear pixels of the input image (Figure 3). The stage of degradation $t$ of these pixels is randomly chosen between 1 and $T$ but applied uniformly to all withheld pixels. This is important because the noise is reduced progressively during inference and the neural network needs to know how to handle intermediate degradation levels.

The loss function is the L2 norm between the actual added noise and the noise predicted by the neural network, computed

over the pixels to which clouds have been added (Figure 4). Pixels which are clouded or covered by land are considered in the last stage of degradation ($T$) during training. Those pixels (in white on panel "added noise (target)" in Figure 4) cannot be used to evaluate the loss function, as the underlying value is not known (for clouded pixels) or not defined (for land pixels).

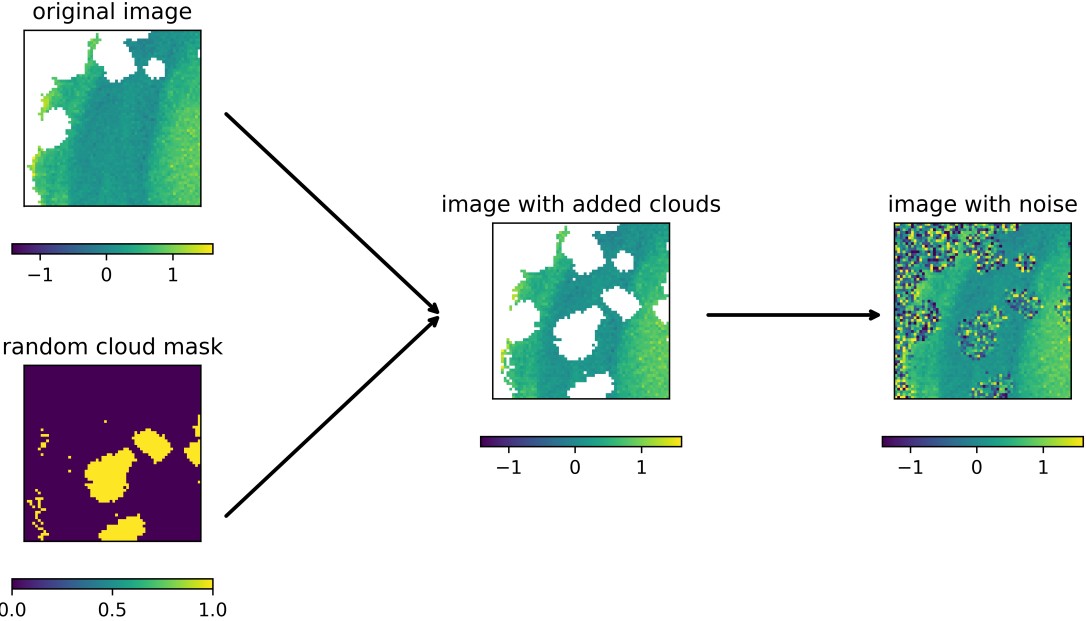

**Figure 3.** Data preparation for training. For the cloud mask, 1 corresponds to a clouded pixel and 0 to a pixel with valid data.

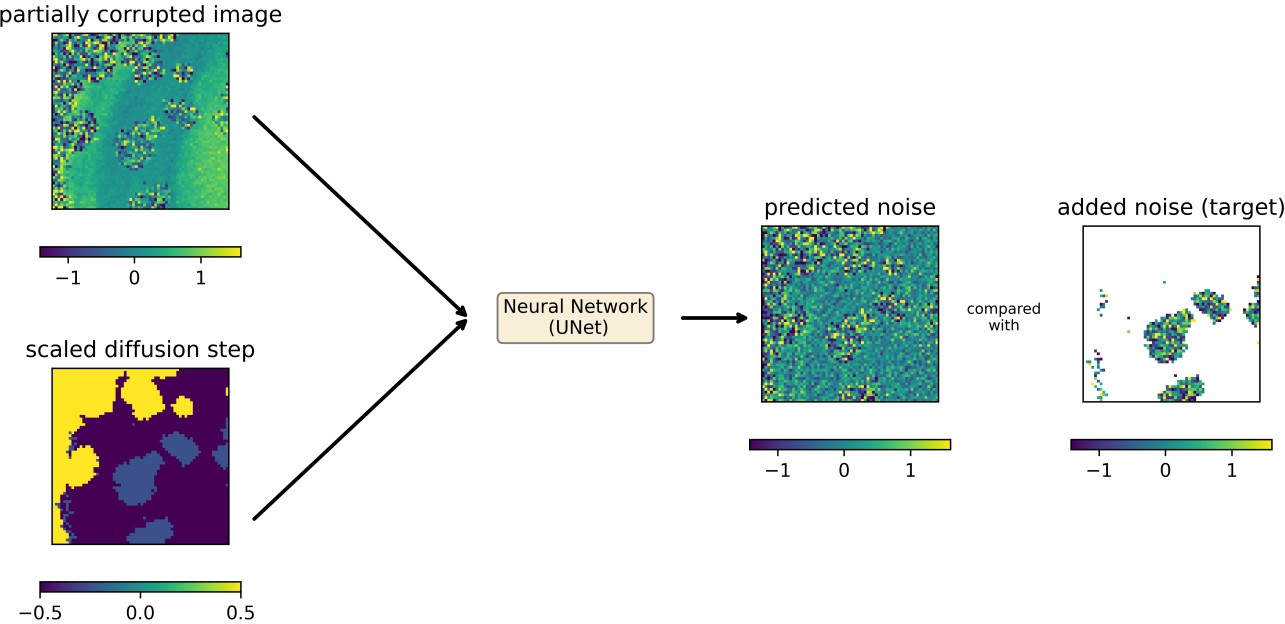

**Figure 4.** Input and output of the neural network during training. Predicted noise is an actual prediction of the trained diffusion model for the provided inputs (units: $\log_{10} \mathrm{mg\,m^{-3}}$). The diffusion step $t$ ($0 \leq t \leq T$) is scaled linearly to the interval $-\frac{1}{2}$ and $\frac{1}{2}$.

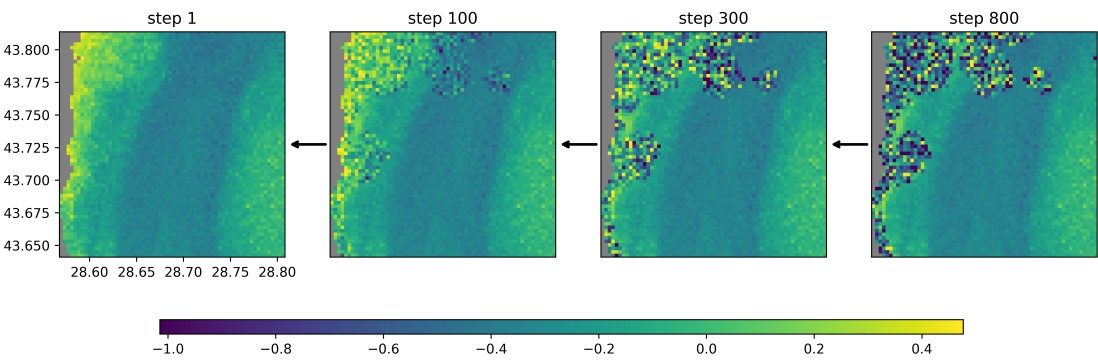

**Figure 5.** Reverse diffusion process illustrated with data from September 9, 2022 (units: $\log_{10} \mathrm{mg\,m^{-3}}$).

The noise schedule of the forward diffusion process is defined by the parameter $\beta_t$, which varies linearly from 0 for $t = 0$ to a maximum value of $\beta_{max}$ for $t = T$, where $\beta_{max}$ and $T$ are hyperparameters (chosen from a search range to satisfy $\bar{\alpha}_T \approx 0$).

The neural network has the general architecture of a U-Net (Ronneberger et al., 2015) which is defined recursively by a block (at a given level $l$) composed of:

- three convolutional layers with output layers $C_l$ and kernel size $k$ followed each by an activation function.

- 2-by-2 max pooling layer.

- inner block at level $l + 1$.

- a single transpose convolution with a stride 2 with the number of output channels the same as the number of input channels of this block followed by an activation function.

- output of the previous layer, added to the input layer to form a residual connection.

An inner block at level $l+1$ has the same structure as an outer block at level $l$, except for the innermost level, where the inner block is simply the identity function. This recursive definition of the U-Net architecture allows us to easily test networks with different depth levels. The depth level $L$, the number of channels $C_l$ ($l = 1 \dots L$) and the kernel size $k$ are hyperparameters of the network.

The input of the neural network is a 2D image with two channels. The first channel is the noisy image (normalized using the mean and standard deviation computed over the training dataset) and a 2D field with the step of the denoising pipeline (scaled between $-\frac{1}{2}$ for clear pixels and $\frac{1}{2}$ for fully-degraded pixels). We do not directly use the step $t$, since the inputs of the neural network should be of the order of 1 to accelerate the training. In this implementation of the denoising diffusion model, every pixel can be at a different step of degradation. During training, noise is intentionally added to the image (advancing from diffusion step $l$ to $l + 1$) and the neural network is trained to predict the noise allowing it to denoise the image and to go from step $l + 1$ back to $l$. The neural network can predict the added noise because it learned the typical spatial structures in the training dataset and it is able to recognise them even in a corrupted image. At a first approximation, the neural network acts like a high-pass filter to identify the noise, which is then removed iteratively during sampling.

The model is optimized using the Adam optimizer (Kingma and Ba, 2014) using the default parameters except for the learning rate. During the training process, the learning rate is repeatedly reduced by a given factor after a certain number of steps. The initial learning rate, the number of steps between the reduction of the learning rate and the reduction factor are treated as hyperparameters.

As usual, all model parameters (weights and biases of all convolutional layers) are optimized using the training data. The denoising diffusion model is implemented in the Julia programming language (Bezanson et al., 2017) using the deep learning library Flux.jl (Innes, 2018; Innes et al., 2018) and the GPU programming library CUDA.jl (Besard et al., 2019, 2018). The training of the neural network takes 7 hours on an NVIDIA A100-SXM4-40GB and 8 hours on an NVIDIA GeForce RTX 4090. The inference time of the test dataset is 30 minutes. All hyperparameters are determined using random search (Bergstra

and Bengio, 2012) to minimize the RMS error of the reconstruction with the validation data (Table 1). The optimal model (in terms of RMS error relative to the validation data) has in total 1.6 million parameters. Unless otherwise stated, all comparisons and reported validation metrics are performed with the independent test data, including the final validation. The final validation is performed using the independent test data.

Table 1. Hyperparameters of the diffusion model with the adopted value and the corresponding search space.

| Parameter | Value | Search space |
|---|---|---|
| kernel size ($k$) | 5 | 3 or 5 |
| channels ($C_l$) | [16, 32, 64, 128] | [16, 32, 64, 128], [16, 32, 64, 128, 256] or [16, 32, 64, 128, 256, 256] |
| activation function | selu | relu, selu or swish |
| number of steps ($T$) | 800 | between 500 and 1500 (step of 100) |
| $\beta_{\max}$ | 0.027 | between 0.01 and 0.04 |
| batch size | 60 | fixed |
| number of epochs | 100 | fixed |
| learning rate | 0.00017 | between $10^{-5}$ and 0.0008 |
| number of epochs before reducing the learning rate | 50 | between 10 and 100 (step of 10) |
| factor by which the learning rate is reduced | 0.938 | between 0.7 and 0.95 |

Preliminary experiments showed that a large training dataset is quite important to obtain a stable reconstruction. In fact, during the reverse diffusion, the neural network is applied 800 times to a satellite image to denoise it and to reconstruct the missing part of the image. Overfitting of the neural network, which emphasizes an unrealistic structure, could quickly lead to an unstable reverse diffusion process (i.e., the variance of the reconstructed image grows in an unbounded way). Such problems were resolved if a sufficiently large and diverse dataset was used for training. In particular, we needed to train the diffusion model using image tiles from the whole Black Sea to obtain a stable reverse diffusion process. As an illustration, a sample of the unconditional generation of images is shown in appendix A together with a random sample of the training data.

## 4.2 Sampling

After training the neural network, the missing data in the validation and test dataset are reconstructed. Every clear pixel of
the input image is considered to be in the non-degraded state $t = 0$, and all other pixels (clouded or on land) are in the fully
degraded state $t = T$ and initialized with normal distributed random values. For these later pixels, the reverse diffusion process
is used iteratively (going from step $l + 1$ to $l$) to reduce their noise keeping the originally present pixels unchanged (Figure
5). The convolution operations in the U-Net ensure spatial coherence between clear pixels and reconstructed pixels. All clear
pixels remain constant during the reverse diffusion because the corresponding term in equation (11) is zero as $\sigma_0 = \sqrt{\beta_0} = 0$
and $\alpha_0 = 1 - \beta_0 = 1$ for these pixels.

For each image of the validation and test two datasets, the reconstruction process is repeated 64 times, leading to an ensemble
of possible reconstructed fields. The larger the ensemble is, the more accurate the derived ensemble mean and variance. Various
ensemble sizes have been used in the literature, for example the ECMWF real-time S2S forecasts use a 51 member ensemble
size (Buizza et al., 2008). Using 64 ensemble members is here a compromise between diversity of ensemble members and
computational time.

From this ensemble, the ensemble mean and the ensemble standard deviation are also computed. When minimizing the RMS
error relative to the validation dataset, only this ensemble mean is considered.

## 5   Results

Figures 6 and 7 show an example of the reconstruction for the dates August 7, 2022 and September 9, 2022 respectively from
the test dataset. In the original data (panel a), additional clouds have been added using the cloud mask from a different image
(panel b) in order to evaluate the accuracy of the reconstruction. From the data with the added clouds, the reverse diffusion
process was performed 64 times. Two of these 64 reconstructions are shown in panels e and f. The ensemble mean (panel c)
and the standard deviation (d) are also computed. For every ensemble member, the interpolated fields in the pixels for which
we have valid values in the input data is, per construction, identical to the initial input value. The ensemble standard deviation
at these locations is thus consequently equal to zero. As expected, the ensemble mean is blurrier at the locations where we
have added clouds, but the individual ensemble members also contain realistic small-scale information at these locations. In
Figure 6 (panel d), we see that the ensemble standard deviation increases near fronts under clouds, since the exact position
of the fronts cannot be deduced from the provided data. In general, the difference between the reconstructions is highest near
the coastline, as the coastal areas are more variable than the offshore waters. This difference is particularly visible when large
clouds are present near the coastline (Figure 7, panels d, e, and f).

Date: 2023-08-07

**Figure 6.** The denoising diffusion model applied on the independent test data for the date 2023-08-07 showing the original data (panel a), the data with added clouds (panel b), the ensemble mean and standard deviation (panel c, d) and two ensemble members (panel e and f). The units are $\log_{10} \mathrm{mg\,m^{-3}}$.

Date: 2022-09-09

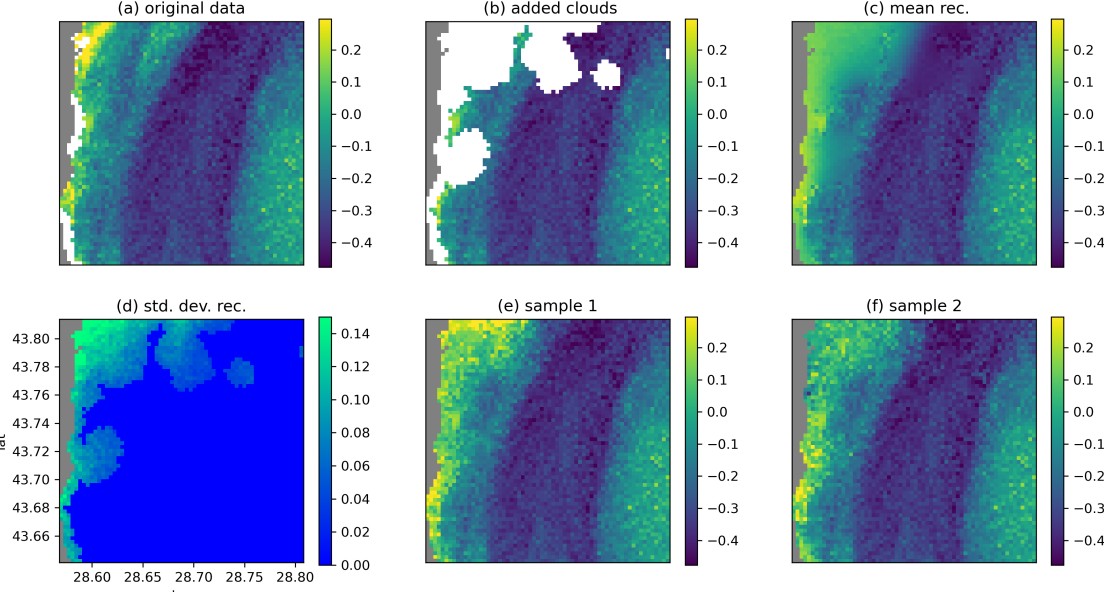

**Figure 7.** The same as Figure 6 for the date September 9, 2022.

We compared the reconstruction with the DINCAE neural network. So far, DINCAE was only trained on data using a fixed
area. We adopted the same approach here and trained DINCAE over the area used for validation. We used the same temporal
split as the diffusion model: data before 2021-09-01 was used for training, the following 12-month period was used to adjust the
hyperparameters (development dataset) and the last 12 months (starting on September 1, 2022) for the independent validation
(test dataset). More information about the application of DINCAE is given in appendix B.

The RMS error and the bias of DINCAE and the diffusion model are computed on artificially clouded pixels for the devel-
opment and test dataset (2 and 3). The RMS error of the diffusion model is based on the ensemble mean. In all cases, the bias
is relatively low and does not contribute significantly to the RMS error. The RMS error of the diffusion model (based on the
ensemble mean) is slightly smaller than the RMS error of DINCAE for development and test datasets. However, as expected
the RMS error of every ensemble member individually is substantially larger than the RMS error of the ensemble mean. Given
that the RMS error is computed over all time instances, the RMS error for a single ensemble member is relatively stable. The
maximum and minimum RMS error among the 64 ensemble members are 0.202 and 0.211 $\log_{10}$ mg m$^{-3}$ respectively.

**Table 2.** Comparison of DINCAE and the diffusion model (using the ensemble mean) with the **development** dataset.

| method | RMS | bias | std(reconstruction) | std(observation) |
|---|---|---|---|---|
| DINCAE | 0.163 | -0.0531 | 0.308 | 0.363 |
| Diffusion Model | 0.151 | 0.00568 | 0.333 | 0.363 |

**Table 3.** Comparison of DINCAE and the diffusion model (using the ensemble mean) with the **test** dataset.

| method | RMS | bias | std(reconstruction) | std(observation) |
|---|---|---|---|---|
| DINCAE | 0.175 | 0.0488 | 0.308 | 0.331 |
| Diffusion Model | 0.163 | 0.00388 | 0.285 | 0.331 |

Figure 8 shows a meandering coastal front with submesoscale flow features, which is partially obscured by the added clouds. The general structure of the front is preserved well by DINCAE and the diffusion model (panels c and e) but the level of details and the intensity is better represented using the diffusion model. The noise visible offshore is retained by the diffusion model (per construction), but it is effectively reduced by DINCAE which can be a desirable effect for some applications.

Date: 2022-11-10

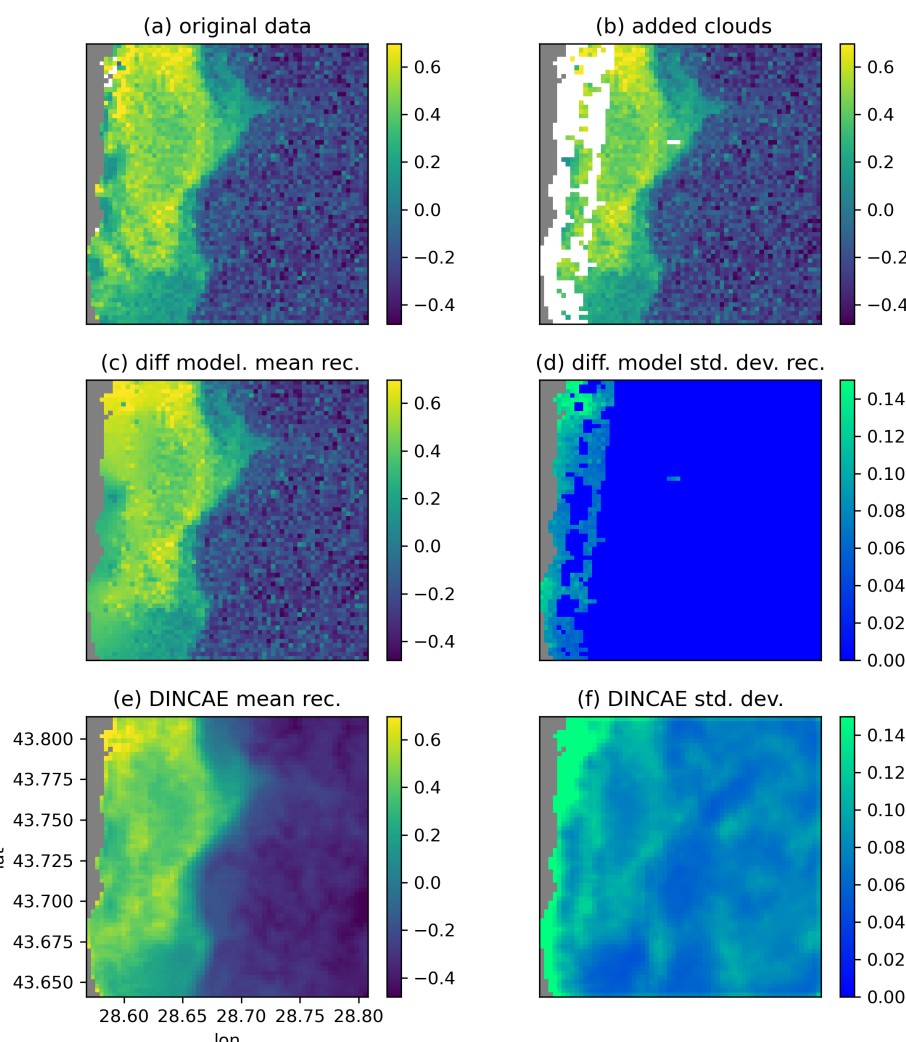

**Figure 8.** Comparison between DINCAE and the diffusion model for the date November 10, 2022 (units: $\log_{10} \text{mg m}^{-3}$).

To assess the scales present in the reconstructed data, a variogram (Cressie, 1991; Wackernagel, 2003) is computed using the reconstruction of the development and test datasets (Figure 9). A variogram of a spatial random field $\phi(\mathbf{x})$ is defined by the following expectation:

$$2\gamma(\mathbf{x}_1, \mathbf{x}_2) = E\left[(\phi(\mathbf{x}_1) - \phi(\mathbf{x}_2))^2\right] \tag{12}$$

Here we are considering a variogram only as a function of distance $h = \|\mathbf{x}_1 - \mathbf{x}_2\|$, which allows us to estimate the variogram numerically by computing the squared differences for the field at randomly chosen locations. These squared differences are averaged over bins of distances using all time instances of the validation and test datasets. As many different random locations were chosen until there are at least 10000 pairs for each bin of distance. For the diffusion model, the variogram is deduced using the individual ensemble members, and the averaging in equation (12) is done also over different ensemble members. When computing the variogram of the original data, only the pairs of points corresponding both to valid pixels are considered.

It can be seen from Figure 9 that both reconstruction methods underestimate the variance in the original data to some degree, but the reconstruction with the diffusion model is consistently closer to the original data than DINCAE, which confirms our qualitative assessment of Figure 8. For the independent test dataset and scales larger than 15 km, the variogram of the diffusion model coincides with the variogram of the original data. The fact that the variogram does not converge to zero as distances tend to zero shows that the data is affected by spatially white noise, as it can be seen in the offshore region of Figure 8 (panel a) which is also called the "nugget effect" (Matheron, 1962). DINCAE effectively removes (or significantly reduces) the spatially uncorrelated white noise and therefore the corresponding variogram shows a clear tendency towards zero for smaller distances.

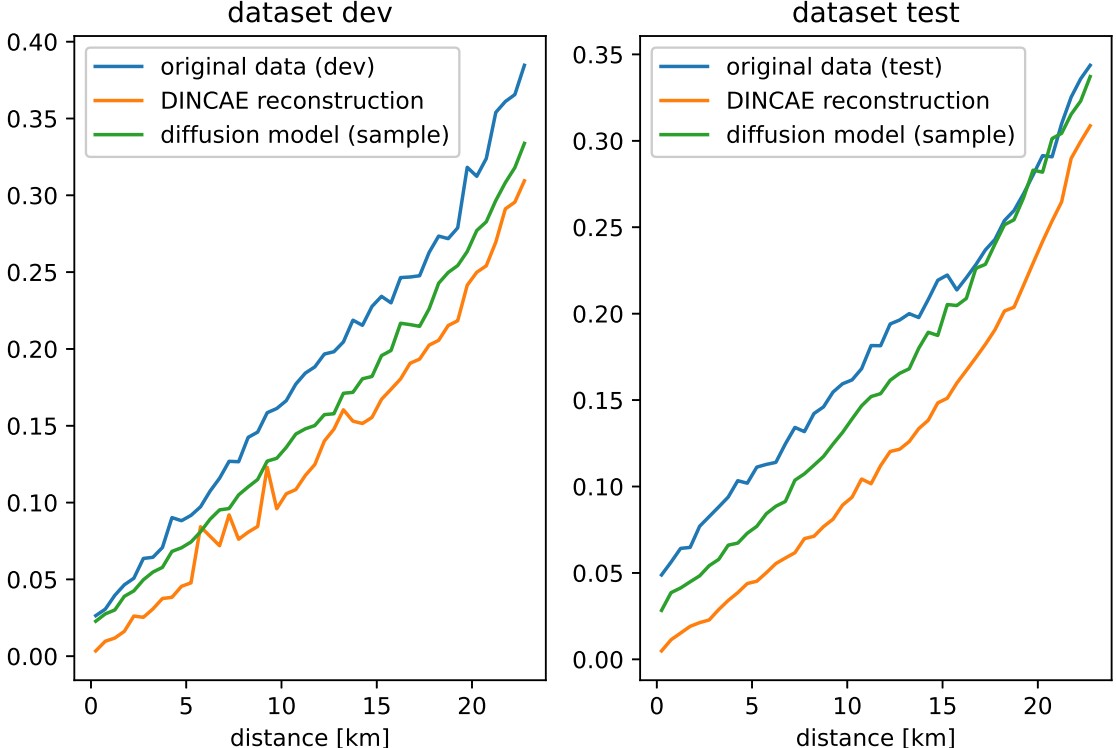

**Figure 9.** Variogram of the development (dev) and test datasets (units: $(\log_{10} \mathrm{mg\,m^{-3}})^2$).

To assess the statistical reliability of the produced reconstruction ensemble, we can use the so-called Talagrand diagram, also called rank histogram (Talagrand et al., 1997; Hamill, 2001). If the ensemble is generated from the same probability distribution as the observations, the ensemble is considered reliable. However, it is important to note that the Talagrand and other statistical tests described below only allow us to assess the reliability of the marginal PDFs (probability density function) evaluated for each pixel individually and not the joint PDF accounting for spatial correlations between pixels.

For each pixel for which an observation is available, the corresponding value of all 64 ensemble members is sorted by $x_1 \le x_2 \cdots \le x_N$ (where here $N = 64$), and the following successive $N+1$ bins are defined as:

$$b_0 = (-\infty, x_1) \tag{13}$$

$$b_i = [x_i, x_{i+1}) \quad \text{for} \ i = 1 \ldots N-1 \tag{14}$$

$$b_N = [x_N, \infty) \tag{15}$$

In this case, the probability that the observations belong to the interval $b_i$ is $\frac{1}{N+1}$ and thus independent of the value of the observation. With a sufficient number of observations, this probability can be estimated for different bins $i$. A Talagrand

diagram shows these frequencies as a function of the bin indices. A perfectly marginally reliable ensemble would result in a flat curve. Underdispersive (or overdispersive) ensembles would result in a ∪-shaped (respectively ∩-shaped) curve.

Figure 10 shows the Talagrand diagram computed for the test for the diffusion model and DINCAE dataset. DINCAE provides the mean and variance of the marginal Gaussian probability distribution function. Therefore, one can derive from this

the corresponding Talagrand diagram using the cumulative distribution function. It can be seen that the error statistics of the diffusion model are closer to the ideal flat curve for the diffusion model than for DINCAE. This shows that the probabilities produced by the diffusion model are marginally reliable, except for the tails of the marginal PDF (first and last bin, corresponding to the probabilities between 1.5% and 98.5%) where the produced ensemble is underdispersive.

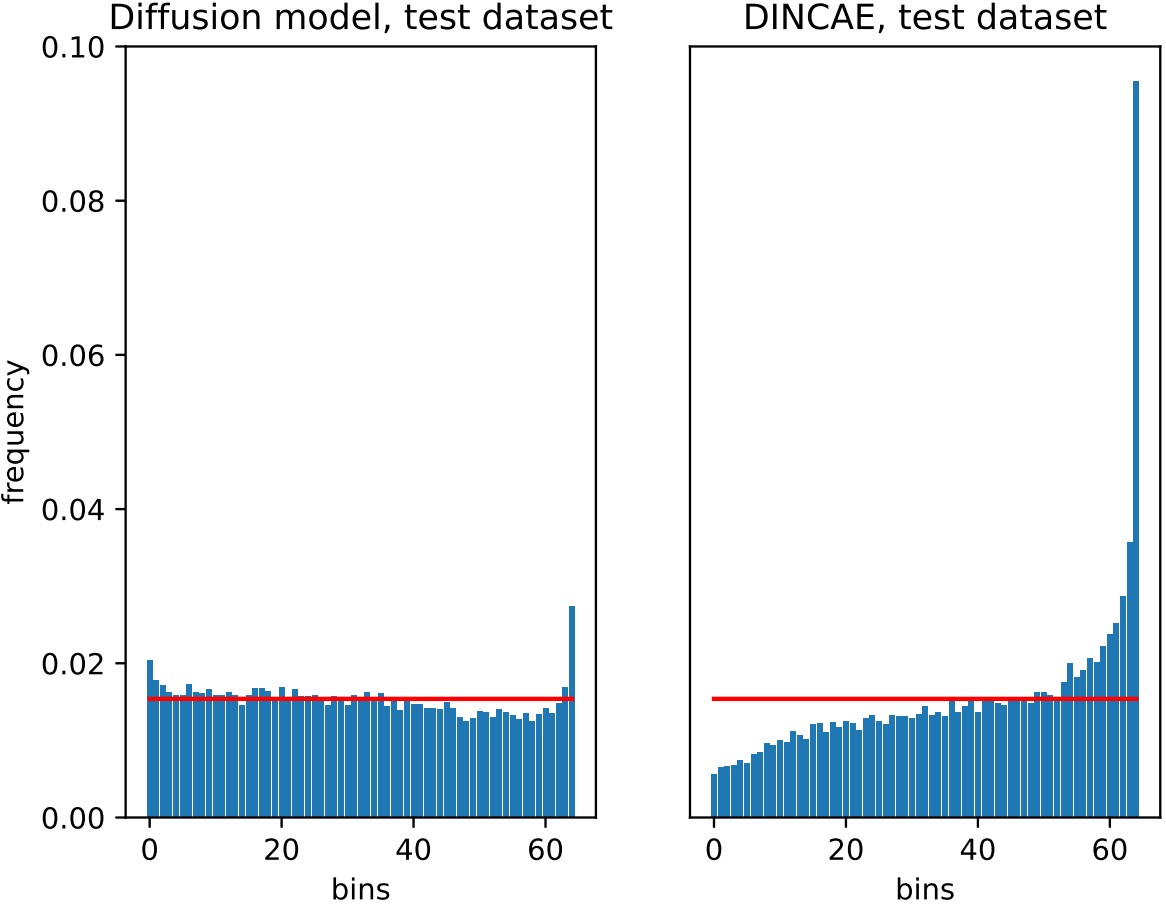

**Figure 10.** Talagrand diagram of the diffusion model and the DINCAE method for the independent test datasets.

Another common probabilistic validation approach defined for marginal PDFs is the Continuous-Ranked Probability Score (CRPS). Following Hersbach (2000), it is defined as:

$$\text{CRPS} = \int_{-\infty}^{\infty} (P(x) - H(x - x_o))^2 dx \tag{16}$$

where $P(x)$ is the cumulative distribution function, $x_o$ the observations, and $H(x)$ the Heaviside function ($H(x) = 1$ for $x \geq 0$ and $H(x) = 0$ otherwise). The CRPS has the same units as the data $x$ and it is always positive or zero. When applied to ensemble reconstructions, the CRPS attains its best score of zero, only when all ensemble members reproduce the observations exactly. The CRPS can be decomposed into potential CRPS (CRPS$_{\text{pot}}$), reliability, uncertainty and resolution:

$$\text{CRPS} = \text{reliability} + \text{CRPS}_{\text{pot}} \tag{17}$$
$$\text{CRPS}_{\text{pot}} = \text{uncertainty} - \text{resolution} \tag{18}$$

The reliability (smaller is better) measures whether the ensemble accurately reflects the uncertainty of the results. Note that a system reproducing the climatological data distribution would be perfectly reliable but would not resolve different events. The resolution (higher is better) determines whether the ensemble allows discrimination between different events. The resolution would be zero for the data climatology. Consequently, the uncertainty is the CRPS score for the data climatology and thus depends only on the variability of the data (and not on the reconstruction method). For more information on these scores and how they are computed based on an ensemble, the reader is referred to Hersbach (2000) and Candille et al. (2007). It should be noted that in this context, the resolution is not related to the spatial or temporal resolution of the dataset.

Table 4 shows the corresponding scores for the test and development datasets and for both considered methods. All scores have the same units of the data and the standard deviation of this training data is 0.46 $\log_{10} \text{mg m}^{-3}$ to provide an order of magnitude of the variability. The reliability of the diffusion model (for the marginal PDF) seems to be quite good, which confirms the results of the Talagrand diagram (Figure 10). The CRPS is mostly determined by the resolution. To further improve the resolution, it might be beneficial to use more data (including multivariate reconstructions), but it is clear that a perfect score is not attainable simply due to the lack of information under clouds.

**Table 4.** Decomposition of the CRPS score using the developpement (dev) and the independent test data for the diffusion model and for DINCAE (units $\log_{10} \text{mg m}^{-3}$).

| method | dataset | CRPS | reliability | CRPS$_{\text{pot}}$ | resolution | uncertainty |
|---|---|---|---|---|---|---|
| diffusion model | dev | 0.0635 | 0.00045 | 0.0631 | 0.130 | 0.193 |
| diffusion model | test | 0.0712 | 0.00041 | 0.0708 | 0.112 | 0.182 |
| DINCAE | dev | 0.0827 | 0.00842 | 0.0743 | 0.119 | 0.193 |
| DINCAE | test | 0.0856 | 0.00356 | 0.0820 | 0.100 | 0.182 |

Among the test data, we took the images with less than 30% of cloud cover (representing 99 images here). To these relatively clear images, we applied the cloud mask (potentially flipped in the longitude or latitude direction) chosen randomly from another image in the test dataset so that the total cloud coverage for every image is within a given range of 45% to 55%. If the cloud coverage is outside this range, then another cloud mask is chosen randomly until the target range is achieved. This procedure is repeated for different ranges, up to a range of 85% to 95% of missing data.

The trained diffusion model was applied to these images, and the RMS error relative to the withheld (and independent) data was computed and is shown in Figure 11.

As expected, the RMS error rises with an increased amount of missing data. With a large amount of missing data, the diffusion model misses the context to reconstruct the field and the model acts as an unconditional diffusion model. It can also be seen that the RMSE does not show any abrupt augmentation.

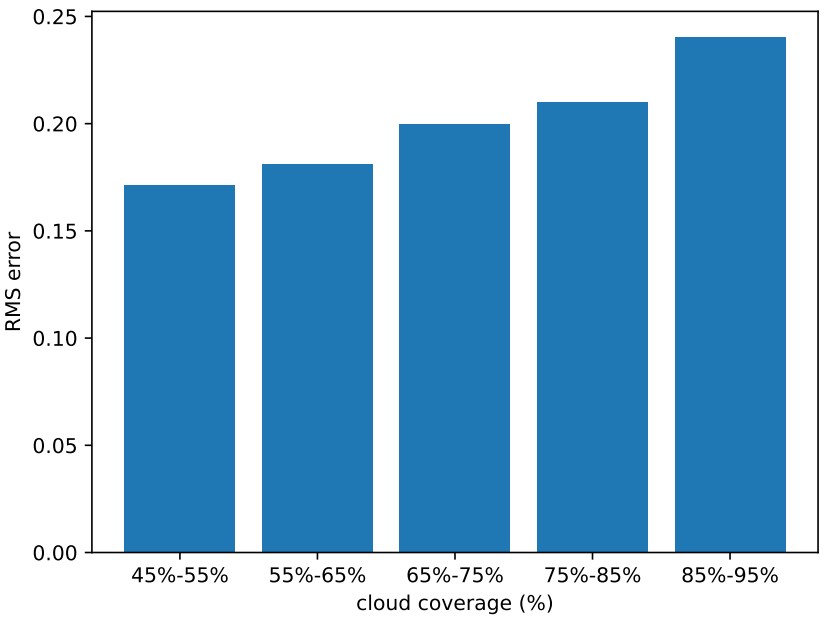

**Figure 11.** Impact of cloud coverage on the RMS computed relative to independent data (units $\log_{10}$ mg m$^{-3}$)

Further domains are considered to test the applicability of the trained diffusion model in comparison with DINCAE to explore the different dynamical regimes. In Figure 12, the domain used previously is labeled as 1, and the additional domains are labeled 2 to 10. For each of these domains DINCAE is trained using only the data from the corresponding domain using the hyperparameters presented in Table B1. As the diffusion model is trained using 64 x 64 tiles from the whole Black Sea, it is not trained again but used only in the inference mode. The RMS error for each domain is shown in table 5 and the corresponding

variogram can be seen in Figure 13. Overall the results from the previous test on the first domain are also applicable to other domains. The RMS error of the diffusion model is lower than the corresponding RMS error of DINCAE except for domain 7. At the same time, the variance for all domains across different scales is more realistic for the diffusion model.

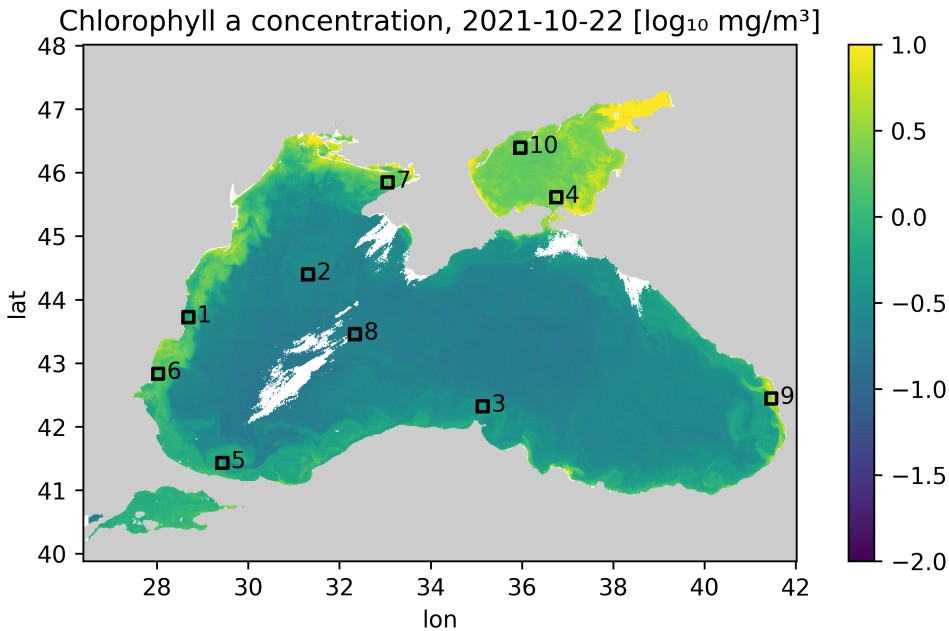

**Figure 12.** Additional domains where the diffusion model is applied (domain 2 to domain 10)

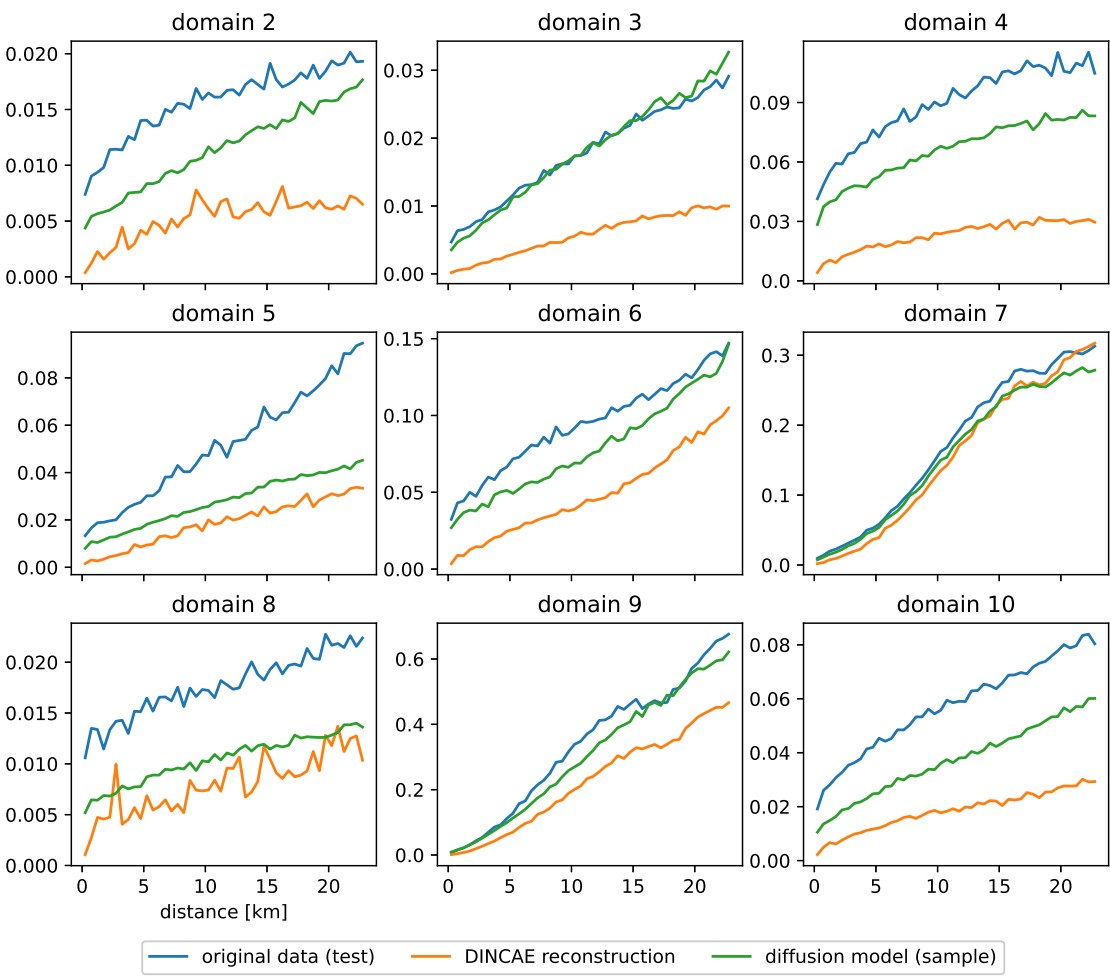

**Figure 13.** Variogram for the independent test data for the additional domains

**Table 5.** RMS error relative to the independent test data for different domains.

| domain | RMS DINCAE | RMS Diffusion Model | std(obs) |
|--------|-----------:|--------------------:|---------:|
| 1      | 0.175      | 0.163               | 0.331    |
| 2      | 0.159      | 0.058               | 0.226    |
| 3      | 0.225      | 0.056               | 0.211    |
| 4      | 0.162      | 0.155               | 0.253    |
| 5      | 0.162      | 0.074               | 0.251    |
| 6      | 0.182      | 0.143               | 0.353    |
| 7      | 0.090      | 0.096               | 0.295    |
| 8      | 0.119      | 0.062               | 0.286    |
| 9      | 0.189      | 0.149               | 0.442    |
| 10     | 0.116      | 0.111               | 0.244    |
| median | 0.158      | 0.107               | 0.289    |

## 6 Conclusions

Denoising diffusion models have shown their great potential for image generation for computer vision applications and related tasks. One limitation of this approach, in the context of satellite data, is that it requires clear images for training. The present manuscript shows that the training approach of Ho et al. (2020) can be extended if the training dataset contains incomplete images. The approach presented here does not need any additional parameters that would require calibration. The spatial coherence and the statistical reliability of the resulting reconstruction process emerges naturally from the training.

The method is tested on relatively small images of the chlorophyll $a$ concentration of the Black Sea. The quality of the reconstruction is assessed using independent test data. The diffusion method compared favorably against the U-Net DINCAE. The RMS error of the reconstructed data using the denoising diffusion model was smaller than the corresponding reconstruction of DINCAE. The main advantage of the diffusion model is however the ability to reproduce an ensemble of possible reconstructed conditions on the available data. Each of these reconstructions contains small-scale information comparable to the scales of variability in the original data, avoiding a common problem where the results of U-Net and autoencoders produce images that are too smooth, as the information on small scales can typically not be recovered under clouds with a certain extent. The overall conclusion is robust when applying this technique to other areas of the Black Sea.

The ensembles of reconstructed data generated by the diffusion model can be used, for example, in the detection of gradients and fronts in the satellite images or in the estimation of the error in derived quantities, where information on how the error is correlated in space is also needed.

Another aspect that would be important to investigate in future studies would be the ability to reconstruct sequences of images, other parameters (like sea surface temperature, salinity, height...), multivariate reconstructions and data with inhomogeneous and/or very reduced coverage like *in situ* observations. It remains to be seen how well the diffusion model can be used in these cases.

*Data availability.* The source code is released as open source under the terms of the MIT License and available at the address https://github.
com/gher-uliege/DINDiff.jl (doi: 10.5281/zenodo.13165363). The satellite chlorophyll $a$ concentration of the Black Sea is provided by the Italian National Research Council (CNR – Rome, Italy) as part of the Copernicus Marine Service (doi: 10.48670/moi-00303).

*Author contributions.* AB designed and implemented the neural network. AB, JB, AAA, BM, CT and JMB contributed to the discussions and to the writing of the manuscript.

*Competing interests.* Aida Alvera Azcárate is a member of the editorial board of the journal Ocean Science.

*Acknowledgements.* The F.R.S.-FNRS (Fonds de la Recherche Scientifique de Belgique) is acknowledged for funding the position of Alexander Barth. The present research benefited from computational resources made available on Lucia, the Tier-1 supercomputer of the Walloon Region, infrastructure funded by the Walloon Region under the grant agreement number 1910247. This work has received funding from the Horizon Europe RIA program via the NECCTON project under the grant agreement number 101081273. Aida Alvera-Azcárate received funding from the Copernicus Marine Service MultiRes project. Copernicus Marine Service is implemented by Mercator Ocean in the frame-
work of a delegation agreement with the European Union. The authors wish also to thank the Julia community, in particular for the Julia programming language and the packages Flux.jl and CUDA.jl.

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

## Appendix A: Sample of training data and generated images

In Figure A1, a random sample of training images are shown. Most training images are affected by a significant amount of noise and some artifacts are present in the training data. The denoising diffusion model aims to generate images with the same distribution and therefore including the noise and artifacts.

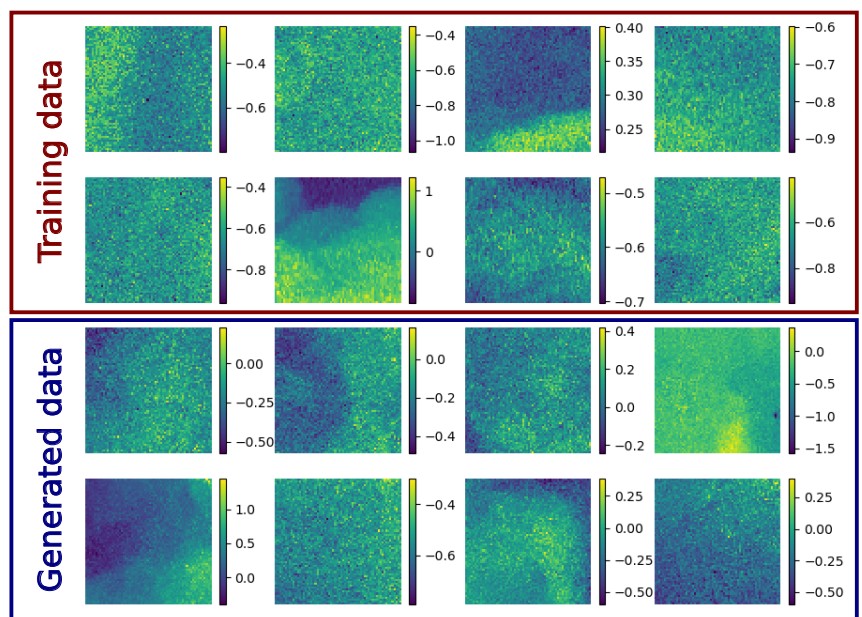

**Figure A1.** Sample of training data and generated images (starting an entirely masked input image)

## Appendix B: Application of DINCAE

As the baseline method, we use the U-Net DINCAE described in Barth et al. (2020) and Barth et al. (2022). The hyperparameters adjusted using the development dataset were the number of epochs, the number of instances in the time window, the upsampling method and whether a refinement step is used. In the case of a refinement step, the neural network is composed of two U-Nets: the first network provides an intermediate estimate of the missing data and the second U-Net uses the intermediate estimate and the original data to provide the final estimate. During training, the loss function is based on a weighted sum of the intermediate and final estimate. For inference, only the final estimate is used. The weights are considered as hyperparameters. More information is provided in Barth et al. (2022).

In Barth et al. (2020), it has been shown that the accuracy of a reconstruction can be improved by averaging the obtained reconstruction over a certain number of epochs after the epoch 200. In practice, we do not save the model weights of the

at different epochs but apply the model on the test and development data and accumulate all the reconstruction which are later normalized to compute the average. The frequency (in number of epochs) of applying the neural network to the test and validation data to compute the corresponding average, is also a hyperparameter here. As before, the hyperparameters were determined by minimizing the RMS error relative to the validation dataset using random search. Table B1 shows all parameters used in DINCAE and their corresponding search range.

The number of parameters of the optimal DINCAE model is 3.1 millions. The training time is 12 minutes on a GeForce RTX 4090 GPU. The inference time of the test and development datasets is 2.7 seconds which is significantly faster than the diffusion model.

**Table B1.** Hyperparameters of DINCAE with the adopted value and the corresponding search space.

| Parameter | Value | Search space |
|---|---|---|
| number of epochs | 1276 | between 500 and 1500 |
| save epochs | 36 | between 10 and 40 |
| batch size | 32 | fixed |
| channels | [32, 64, 128, 256, 512] | fixed |
| instances in time window | 1 | 1, 3 or 5 |
| upsampling method | nearest | nearest or bilinear |
| refinement step | deactivated | activated or deactivated |