# Peer review of "Ensemble reconstruction of missing satellite data using a denoising diffusion model: application to chlorophyll a concentration in the Black Sea"

_EGUsphere, 2024_

## Referee Comment (RC1)

**Review of the manuscript titled: Ensemble reconstruction of missing satellite data using a denoising diffusion model: application to chlorophyll $a$ concentration in the Black Sea**

**May 2024**

The manuscript titled "Ensemble reconstruction of missing satellite data using a denoising diffusion model: application to chlorophyll $a$ concentration in the Black Sea" presents a novel method for the reconstruction of missing data in satellite observations. The method is based on the denoising diffusion probabilistic model, where a sample form a noise distribution is incrementally transformed to a sample from the target distribution. In this case, the authors use this approach to fill-in missing patches in the satellite observations, with the aim of: creating a spatially coherent reconstruction, and to create an ensemble of reconstructions, which better describe data variations compared to a single mean estimator for the missing value. At each step in the noise reduction process, the authors use a U-net style neural network to predict the necessary noise delta, which, if applied to the the current step input, reduces the total amount of noise in the image, approaching the target reconstruction. The authors test their method on the task of reconstructing chlorophyll $a$ values for a select region over the Black Sea, with good results implying, that the method embodies a meaningful addition to the roster of spatial data reconstruction techniques.

**1 General comments**

Overall, the manuscript is structured well and clearly demonstrates the application of denoising diffusion probabilistic models in the domain of satellite reconstruction, with compelling results when compared to the baseline method, denoted as DINCAE (a method which was previously applied on this task). However, before I can fully recommend the manuscript for publication, there are some shortcomings that have to be addressed.

Firstly, there are some implementation details which might indicate potential errors in the algorithm's implementation. It seems that the presented equations pertaining to the diffusion model somewhat deviate from the standard definition, while the changes lack an explanation or motivation.

Secondly, the results section, although demonstrating that the proposed method compares favourably to the baseline approach in terms of RMSE, variogram, and quality of reconstruction, could still benefit from some additional comparisons. Furthermore, I do not wholly agree with some of the conclusions reached by the authors with regards to the interpretation of the Talagrand diagram.

I provide detailed arguments for each of the issues raised in the following Section.

**2 Specific comments**

**2.1 Denoising diffusion probabilistic model implementation**

- Equation (3): The authors state that the conditional distribution of the image $x$ at step $t$ given $x_0$ in the forward pass is defined as $q(x_t|x_0) = \mathcal{N}(x_t; \sqrt{\bar{\alpha}_t}x_0, \bar{\alpha}_t\boldsymbol{I})$. This suggests that the value of the variance approaches zero as $t$ increases, reducing the distribution to be zero mean and zero standard deviation in the limit. The value of the variance in the conditional case, if I am not mistaken, should be equal to $(1 - \bar{\alpha}_t)\boldsymbol{I}$, given the transformation defined by Equation (1).

- Equation (9): The reverse probability $p_\Theta(x_T)$ should not be parameterised by $\Theta$, since the distribution is defined in Equation (5), where no such parameters are present. If these distributions indeed differ the authors should explain what properties the parameterization defines in this specific case. Additionally, a technical mistake seems to be present in the term $q(x_{1:T})|x_0$, which should be equal to $q(x_{1:T}|x_0)$ correct?

**2.2 Diffusion model neural network description**

This comment pertains to the neural network description provided in paragraphs 185 and 190, and Table 1. The definition of the neural network is given recursively, with each block $l$ being dependent on the block $l-1$. However, given how the levels are provided $l = 1, ..., L$ ($L$ being the depth level) and $C_l = [16, 32, 64, 128]$, this description might be confusing for readers unfamiliar with the architecture. For example, one can make the mistake that the number of channels on the first level $C_1$ is equal to 16. However, as far as I understand the provided description, the block at depth level 4 contains 16 channels while the block at depth level 1 contains 128 channels. Therefore, the initial block corresponds to $l = 4$, while the "deepest" block corresponds to $l = 1$, which seem counter-intuitive given that $l$ denotes the depth level.

Consider these two cases: if $l = 1$ and $C_1 = 16$, then the inner block of block $l = 1$ is an identity and the recursion stops immediately. However, if one assumes that $l = 4$ and $C_4 = 16$, then the inner block at $l = 3$ contains $C_3 = 64$ etc. which results in the familiar U-net architecture, where the spatial dimension is reduced with each consequent block and the number of channels increases. This later assumption is not self evident from the provided description. Therefore, I suggest that the author either flip the depth indices $l$, such that $l = L, ..., 1$, or that they flip the $C_l$ values $C_l = [128, 64, 32, 16]$ while keeping the indices intact.

**2.3 Description of the DINCAE method**

The authors provide a short description of the DINCAE method's training setup in the Results section, in paragraphs 250 and 255. While I believe that this is beneficial to the manuscript the description somewhat breaks the flow of the Results section. Therefore, I suggest that the authors move this description to the Appendix.

**2.4 Talagrand diagram in Figure 10**

The authors compute the Talagrand diagram using the ensemble as an approximate distribution function, where each ensemble memeber represents an equally probable event realization. The authors sort the ensemble members in an ascending order for each masked pixel, independently. The resulting empirical distribution functions and their corresponding ground truth values are used to construct the diagram. The histograms for both the *dev* and *test* datasets are displayed in Figure 10 and the authors conclude that: "*Figure 10 shows the Talagrand diagram computed for the development and testing dataset. It can be seen that except for the two first and two last bins (corresponding to the probabilities between 3% and 97%), the Talagrand diagram is relatively flat. This shows that the produced probabilities are reliable, except for very rare events where the produced ensemble is underdispersive. The difficulty of predicting rare events is a known issue in machine learning (e.g. Kaiser et al., 2017) and a dedicated area of research.*"

Here I would like to raise a minor concern regarding the use of "*probabilities are reliable*" in this context. The produced probabilities are *marginally reliable*, since each pixel is treated independently from its neighbours. However, this does not necessarily imply that the joint distribution of the ensemble is reliable, which is not decernable from the conclusion reached by the authors. For example, consider taking the same ensemble forecast produced in this work, however, with its values randomly permuted between the corresponding members for each pixel. A permuted forecast like this would exhibit the same Talagrand diagram (since the sorting on a per-pixel basis restores the initial diagram conditions), however, the forecast would not be jointly reliable as the spatial relationships would be lost. Therefore, I suggest that the authors state that this evaluation method, as is, assesses the marginal reliability only and not the joint. Again, this is not a major concern for readers familiar with the evaluation technique, however, since the spatial correlation is an important asset of this proposed reconstruction method, a clarification of this would be welcome.

However, I do not agree that the excess number of observations in the extreme ranks implies that the method perform poorly for very rare events only. The excess denotes that the distribution described by the ensemble exhibits short tails, meaning, that a disproportionate number of observations fall into those ranks. These observations can including realizations that are not rare at all and should actually be described by other ranks. Therefore, I would suggest rewording this conclusion such that it reflect the notion of the distribution tails being too short rather than an explicit comment on the reliability of rare event forecasting.

**2.5   DINCAE comparisons**

The output of the DINCAE method can be interpreted as a normal distribution, where the reconstruction is its mean, and the reconstruction error its standard deviation (or variance), correct? If so, I suggest constructing the Talagrand diagram for the DINCAE method as well, which would further demonstrate the impact of the proposed method's distributional capabilities compared to the baseline. The same comment applies to the evaluation using the CRPS method, where the DINCAE approach (given that the above assumption holds) can also be included.

I also suggest that the authors include the training and inference times for the DINCAE method, as well as the number of parameters of the DINCAE method, such that the reader can better asses the relative computational complexity of this new approach compared to the baseline.

**2.6   Diffusion model performance conditional on the number of valid input image pixels**

The proposed diffusion model is dependent on the valid pixel (pixels without clouds) in the input image to construct a spatially consistent reconstruction. This approach produces realistic reconstructions with a high degree of spatial correlation as can be seen in the provided examples. This, however, prompted the following consideration: how does the performance of the reconstruction degrade in relation to the number of valid pixel available in the input image? An evaluation like this could be an interesting inclusion in the current manuscript, providing a practitioner with the knowledge on how reliable the reconstruction is given how much information is present in the original input image. The ensemble spread might already describe such notions however, it might not be marginally reliable when considering images with a high degree of missing valid data.

**2.7   Miscellaneous comments**

- On "*For the validation and test data, we randomly took the cloud mask from other time instances to mask additional grid cells which will be used for validation. Only images with a cloud mask between 15% and 35% of the missing date were considered as an additional mask to obtain a sufficient number of "clouded" pixels without masking an image almost entirely.*" in paragraph 150: Does this mean that, when constructing an input image from the validation/testing datasets, a random image with 15% to 30% of missing data is selected (still from the same dataset) and its cloud mask is used to cover the current input image's pixels? If so, it seems that this approach can still result in a completely covered image if the image being masked has a coverage greater than 85%, correct? Or are only images with a coverage of less than 70% considered for the validation/test datasets? A few comments on this would be appreciated.

- On "*During training, for each image of the training dataset, a different image is randomly selected (also from the training dataset) and its cloud mask is used to degrade clear pixels of the input image (Figure 3). The stage of degradation t of these pixels is randomly chosen between 1 and T.*" in paragraph 170: Can it not occur that the training image can be fully degraded after the additional cloud mask is provided (example: input image has 20% valid data and the mask has a cover of 80%)? Such training images might slow the convergence of the method as the denoising process is completely unguided. Or is this event rare in practice?

  Furthermore, what is the benefit of setting the degradation value between 1 and $T$ instead of just $T$? If I understand correctly, during inference, each missing pixel is treated as being fully degraded. Is there

a difference in performance compared to setting all pixels to the fully degraded value $T$? Does this help in cases where the training image might be degraded to a high spatial degree (above example)? A few comments on this would be appreciated.

- On "*For each image of the validation and test two datasets, the reconstruction process is repeated 64 times, leading to an ensemble of possible reconstructed fields.*" in paragraph 230: How did you choose the number of ensemble members (64 members) in the reconstruction? Was it determined empirically? If so, please provide an explanation.

- On "*In Barth et al. (2020), it has been shown that the accuracy of a reconstruction can be improved by averaging the obtained reconstruction over a certain number of epochs after the epoch 200.*" in paragraph 250: I do not fully understand this approach. Does this mean that, during training, intermediate models from epoch 200 onwards are saved and the mean reconstruction from each of those models is used as the final DINCAE output?

**3   Technical comments**

- Figure 2, Figure 6, Figure 7, Figure 8: Consider adding lat, lon labels to the axis.

- Broken Latex mathematical symbol for $\bar{\alpha}_T$ in paragraph 180.

- The kernel size $k_s$ (Table 1) does not require a subscript since it is a fixed value across levels. Consider omitting the subscript.

- "As an illusion" in paragraph 215: misplaced use of the word "illusion". Consider replacing with "illustration".

- Table 2: Typo in "desactivated". Additionally, consider explaining the meaning of the rows of the table as some are not self evident, for example "refinement step".

- On "*(corresponding to the probabilities between 3% and 97%)*": Should this not be equal to "between 1.5% and 98.5%" since each interval has a weight of $\frac{1}{65}$ implying, that the first rank covers realizations with a probability of occurrence between 0 and 0.015 and the last rank between 0.98 and 1? Therefore, the middle ranks exhibit a coverage between 0.015 and 0.98, correct?

---

## Author Comment (AC1)

**1 Reviewer 1**

**1.1 General Comments**

Overall, the manuscript is structured well and clearly demonstrates the application of denoising diffusion probabilistic models in the domain of satellite reconstruction, with compelling results when compared to the baseline method, denoted as DINCAE (a method which was previously applied on this task). However, before I can fully recommend the manuscript for publication, there are some shortcomings that have to be addressed. Firstly, there are some implementation details which might indicate potential errors in the algorithm's implementation. It seems that the presented equations pertaining to the diffusion model somewhat deviate from the standard definition, while the changes lack an explanation or motivation. Secondly, the results section, although demonstrating that the proposed method compares favourably to the baseline approach in terms of RMSE, variogram, and quality of reconstruction, could still benefit from some additional comparisons. Furthermore, I do not wholly agree with some of the conclusions reached by the authors with regards to the interpretation of the Talagrand diagram. I provide detailed arguments for each of the issues raised in the following Section.

We thank the reviewer for her/his careful reading of the manuscript. Essentially, we agree with the proposed changes and implement them in the updated manuscript. Unfortunately, there were some typos in the original manuscript. However, these typos only affected the presentation and not the implementation. The source code of the diffusion model has also been made available (https://github.com/gher-uliege/DINDiff.jl). As proposed by the reviewer, we extended the discussion of the results (in particular, computing the Talagrand diagram and the CRPS for the DINCAE method). The interpretation of the Talagrand diagram was also updated. More information is given in the point-by-point response below.

**1.2 Specific Comments**

Equation (3): The authors state that the conditional distribution of the image x at step t given x0 in the forward pass is defined as $q(\mathbf{x}_t|\mathbf{x}_0) = N(\mathbf{x}_t; \sqrt{\bar{\alpha}_t}\mathbf{x}_0, \alpha_t\mathbf{I})$ This suggests that the value of the variance approaches zero as $t$ increases, reducing the distribution to be zero mean and zero standard deviation in the limit. The value of the variance in the conditional case, if I am not mistaken, should be equal to $(1 - \bar{\alpha}_t)\mathbf{I}$, given the transformation defined by Equation (1).

The reviewer is, of course, correct. Thank you for spotting this issue, which is corrected in the revised manuscript. The equation now reads:

$$q(\mathbf{x}_t|\mathbf{x}_0) = N(\mathbf{x}_t; \sqrt{\bar{\alpha}_t}\mathbf{x}_0, (1 - \bar{\alpha}_t)\mathbf{I}) \tag{1}$$

Fortunately, Equation 3 is not used in the code implementation, only its limit (equation 4) is used, which is not affected by this error.

Equation (9): The reverse probability $p_{\boldsymbol{\theta}}(x_T)$ should not be parameterised by $\theta$, since the distribution is defined in Equation (5), where no such parameters are present. If these distributions indeed differ the authors should explain what properties the parameterization defines in this specific case. Additionally, a technical mistake seems to be present in the term $q(\mathbf{x}_{1:T})|\mathbf{x}_0$, which should be equal to $q(\mathbf{x}_{1:T}|\mathbf{x}_0)$ correct?

The review is certainly correct. Thank you for finding these issues. The updated equation now reads:

$$E\left[-\log(p_{\boldsymbol{\theta}}(\mathbf{x}_0))\right] \leq -E\left[\log \frac{p_{\boldsymbol{\theta}}(\mathbf{x}_{0:T})}{q(\mathbf{x}_{1:T}|\mathbf{x}_0)}\right] = -E\left[\log\left(p(\mathbf{x}_T)\prod_{t=1}^{T}\frac{p_{\boldsymbol{\theta}}(\mathbf{x}_{t-1}|\mathbf{x}_t)}{q(\mathbf{x}_t|\mathbf{x}_{t-1})}\right)\right] = L_{\text{elb}} \qquad (2)$$

This comment pertains to the neural network description provided in paragraphs 185 and 190, and Table 1. The definition of the neural network is given recursively, with each block $l$ being dependent on the block $l - 1$. However, given how the levels are provided $l = 1, \ldots, L$ ($L$ being the depth level) and $C_l = [16, 32, 64, 128]$, this description might be confusing for readers unfamiliar with the architecture. For example, one can make the mistake that the number of channels on the first level $C_1$ is equal to 16. However, as far as I understand the provided description, the block at depth level 4 contains 16 channels while the block at depth level 1 contains 128 channels. Therefore, the initial block corresponds to $l = 4$, while the "deepest" block corresponds to $l = 1$, which seem counter-intuitive given that l denotes the depth level. Consider these two cases: if $l = 1$ and $C_1 = 16$, then the inner block of block $l = 1$ is an identity and the recursion stops immediately. However, if one assumes that $l = 4$ and $C_4 = 16$, then the inner block at $l = 3$ contains $C_3 = 64$ etc. which results in the familiar U-net architecture, where the spatial dimension is reduced with each consequent block and the number of channels increases. This later assumption is not self evident from the provided description. Therefore, I suggest that the author either flip the depth indices $l$, such that $l = L, ..., 1$, or that they flip the $C_l$ values $C_l = [128, 64, 32, 16]$ while keeping the indices intact.

We are sorry for the confusion. Indeed, the recursive definition should be reversed relative to the depth index $l$. In particular, at line 187 (submitted version)

inner block at level $l - 1$

Should rather be:

inner block at level $l + 1$.

Similar changes have been made at the lines 191-194. The input resolution is indeed at level $l = 1$ (with 16 channels) and the deepest levels $l = 4$ (with 128 channels). We verified that this is actually the case in the code implementation used (i.e. number of channels increases as spatial dimensions decrease) .

The authors provide a short description of the DINCAE method's training setup in the Results section, in paragraphs 250 and 255. While I believe that this is beneficial to the manuscript the description somewhat breaks the flow of the Results section. Therefore, I suggest that the authors move this description to the Appendix.

We agree, and the relevant text and the hyperparameters of DINCAE is now in the appendix.

**Talagrand diagram in Figure 10**

The authors compute the Talagrand diagram using the ensemble as an approximate distribution function, where each ensemble memeber represents an equally probable event realization. The authors sort the ensemble members in an ascending order for each masked pixel, independently. The resulting

empirical distribution functions and their corresponding ground truth values are used to construct the diagram. The histograms for both the dev and test datasets are displayed in Figure 10 and the authors conclude that: "Figure 10 shows the Talagrand diagram computed for the development and testing dataset. It can be seen that except for the two first and two last bins (corresponding to the probabilities between 3% and 97%), the Talagrand diagram is relatively flat. This shows that the produced probabilities are reliable, except for very rare events where the produced ensemble is underdispersive. The difficulty of predicting rare events is a known issue in machine learning (e.g. Kaiser et al., 2017) and a dedicated area of research."

Here I would like to raise a minor concern regarding the use of "probabilities are reliable" in this context. The produced probabilities are marginally reliable, since each pixel is treated independently from its neighbours. However, this does not necessarily imply that the joint distribution of the ensemble is reliable, which is not decernable from the conclusion reached by the authors. For example, consider taking the same ensemble forecast produced in this work, however, with its values randomly permuted between the corresponding members for each pixel. A permuted forecast like this would exhibit the same Talagrand diagram (since the sorting on a per-pixel basis restores the initial diagram conditions), however, the forecast would not be jointly reliable as the spatial relationships would be lost. Therefore, I suggest that the authors state that this evaluation method, as is, assesses the marginal reliability only and not the joint. Again, this is not a major concern for readers familiar with the evaluation technique, however, since the spatial correlation is an important asset of this proposed reconstruction method, a clarification of this would be welcome.

However, I do not agree that the excess number of observations in the extreme ranks implies that the method perform poorly for very rare events only. The excess denotes that the distribution described by the ensemble exhibits short tails, meaning, that a disproportionate number of observations fall into those ranks. These observations can including realizations that are not rare at all and should actually be described by other ranks. Therefore, I would suggest rewording this conclusion such that it reflect the notion of the distribution tails being too short rather than an explicit comment on the reliability of rare event forecasting.

The review is certainly right that the Talagrand and other statistics only test if the probabilities are *marginally* reliable. We clarified this in the revised manuscript and changed "reliable" by "marginal reliable" at several places in this section.

We also followed the reviewer's suggestion and removed reference to "rare" events and made explicit reference to the tails of the underlying PDF. The relevant sentence now reads:

If the ensemble is generated from the same probability distribution as the observations, the ensemble is considered reliable. However, it is important to note that the Talagrand and other statistical tests described below only allow us to assess the reliability of the marginal PDFs (probability density function) evaluated for each pixel individually and not the joint PDF accounting for spatial correlations between pixels. [...] This shows that the produced probabilities are marginally reliable, except for the tails of the marginal PDF where the produced ensemble is underdispersive.

**DINCAE comparisons**

> The output of the DINCAE method can be interpreted as a normal distribution, where the reconstruction is its mean, and the reconstruction error its standard deviation (or variance), correct? If so, I suggest constructing the Talagrand diagram for the DINCAE method as well, which would further demonstrate the impact of the proposed method's distributional capabilities compared to the baseline. The same comment applies to the evaluation using the CRPS method, where the DINCAE approach (given that the above assumption holds) can also be included.
>
> I also suggest that the authors include the training and inference times for the DINCAE method, as well as the number of parameters of the DINCAE method, such that the reader can better asses the relative computational complexity of this new approach compared to the baseline.

The reviewer is correct that the DINCAE method provides a mean and standard deviation for every pixel, and the marginal PDFs are treated as a Gaussian distribution.

In the revised manuscript, we added the Talagrand diagram and the CRPS statistics (and its decomposition) for the DINCAE method, as they all rely only on marginal distributions, as pointed out by the reviewer. For DINCAE, the Talagrand diagram was constructed using the Gaussian cumulative distribution function, while for the CRPS statistics, we created an ensemble with 10 000 samples following the marginal PDF.

The corresponding table and figures have been revised in the new manuscript and show that the diffusion model is more reliable (assessing the marginal PDFs) than DINCAE.

The number of parameters of the optimal DINCAE model is 3.1 millions. The training time is 12 minutes on a GeForce RTX 4090 GPU. The inference time of the test and development datasets is 2.7 seconds which is significantly faster than the diffusion model.

> **Diffusion model performance conditional on the number of valid input image pixels**
>
> The proposed diffusion model is dependent on the valid pixel (pixels without clouds) in the input image to construct a spatially consistent reconstruction. This approach produces realistic reconstructions with a high degree of spatial correlation as can be seen in the provided examples. This, however, prompted the following consideration: how does the performance of the reconstruction degrade in relation to the number of valid pixel available in the input image? An evaluation like this could be an interesting inclusion in the current manuscript, providing a practitioner with the knowledge on how reliable the reconstruction is given how much information is present in the original input image. The ensemble spread might already describe such notions however, it might not be marginally reliable when considering images with a high degree of missing valid data.

We agree and have added the following additional test to the manuscript.

Among the test data, we took the images with less than 30% of cloud cover (representing 99 images here). To these relatively clear images, we applied the cloud mask (potentially flipped in the longitude or latitude direction) chosen randomly from another image in the test dataset so that the total cloud coverage for every image is within a given range of 45% to 55%. If the cloud coverage is outside this range, then another cloud mask is chosen randomly until the target range is achieved. This procedure is repeated for different ranges, up to a range of 85% to 95%

of missing data.

The trained diffusion model was applied to these images, and the RMS error relative to the withheld (and independent) data was computed and is shown in Figure 1.

As expected, the RMS error rises with an increased amount of missing data. With a large amount of missing data, the diffusion model misses the context to reconstruct the field and the model acts as an unconditional diffusion model. It can also be seen that the RMSE does not show any abrupt augmentation.

[Figure]

Figure 1: Impact of cloud coverage on the RMS computed relative to independent data (units $\log_{10}\mathrm{mg\ m^{-3}}$)

On "For the validation and test data, we randomly took the cloud mask from other time instances to mask additional grid cells which will be used for validation. Only images with a cloud mask between 15% and 35% of the missing date were considered as an additional mask to obtain a sufficient number of "clouded" pixels without masking an image almost entirely." in paragraph 150: Does this mean that, when constructing an input image from the validation/testing datasets, a random image with 15% to 30% of missing data is selected (still from the same dataset) and its cloud mask is used to cover the current input image's pixels? If so, it seems that this approach can still result in a completely covered image if the image being masked has a coverage greater than 85%, correct? Or are only images with a coverage of less than 70% considered for the validation/test datasets? A few comments on this would be appreciated.

The reviewer is right, and the procedure, while not likely, could in theory result in a completely masked image. However, we checked that this is not the case for the generated dataset. The following has been added to the manuscript to clarify this point:

Only images with a cloud mask between 15% and 35% of the missing data were considered as an additional mask to obtain a sufficient number of "clouded" pixels and to reduce the risk that an image is masked almost entirely. We verified that, neither in the validation nor in the test dataset, were images masked entirely after applying the cloud mask.

On "During training, for each image of the training dataset, a different image is randomly selected (also from the training dataset) and its cloud mask is used to degrade clear pixels of the input image (Figure 3). The stage of degradation t of these pixels is randomly chosen between 1 and T." in paragraph 170: Can it not occur that the training image can be fully degraded after the additional cloud mask is provided (example: input image has 20% valid data and the mask has a cover of 80%)? Such training images might slow the convergence of the method as the denoising process is completely unguided. Or is this event rare in practice?

We agree that this can happen and would slow down the training process, as the reviewer points out. For the training data, we estimated this probability numerically (using 100 000 000 pairs of images chosen at random) and found that the probability is 0.00071289. It is indeed a quite rare event.

Furthermore, what is the benefit of setting the degradation value between 1 and T instead of just T? If I understand correctly, during inference, each missing pixel is treated as being fully degraded. Is there a difference in performance compared to setting all pixels to the fully degraded value T? Does this help in cases where the training image might be degraded to a high spatial degree (above example)? A few comments on this would be appreciated.

During inference, each missing pixel is only treated as fully degraded *initially* at step $T$. After removing the noise predicted by the neural network, those pixels will be at the step $T - 1$. For inference, we need to apply the neural network multiple times (here $T = 800$) to reach the clear and non-degraded level ($t = 0$). So, the neural network needs to know how to handle intermediate degradation levels during inference. This information has been added to the manuscript.

On "For each image of the validation and test two datasets, the reconstruction process is repeated 64 times, leading to an ensemble of possible reconstructed fields." in paragraph 230: How did you choose the number of ensemble members (64 members) in the reconstruction? Was it determined empirically? If so, please provide an explanation.

We did not test different ensemble sizes. The number is rather guided by the typical ensemble size used in ensemble modeling in oceanography (*e.g.* Simon and Bertino, 2009; Ohishi et al., 2022) and meteorology (Buizza et al., 2008). In theory, the method should work better as the ensemble size increases towards infinity. If we had chosen too large ensemble sizes, one could have criticized the method as having only been tested in an impracticable setting.

On "In Barth et al. (2020), it has been shown that the accuracy of a reconstruction can be improved by averaging the obtained reconstruction over a certain number of epochs after the epoch 200." in paragraph 250: I do not fully understand this approach. Does this mean that, during training, intermediate models from epoch 200 onwards are saved and the mean reconstruction from each of those models is used as the final DINCAE output?

Yes, the reviewer is correct that this is essentially the approach employed here. It is similar to ensemble averaging different trained models, except that we use the same model at different epochs, which does not increase the computational costs of the training. In practice, we do not save the model weights at different epochs but apply the model to the test and development data and accumulate all the reconstructions, which are later normalized to compute the average.

**1.3 Technical comments**

Figure 2, Figure 6, Figure 7, Figure 8: Consider adding lat, lon labels to the axis.

Done!

Broken Latex mathematical symbol for $\bar{\alpha}_T$ in paragraph 180.

Fixed!

The kernel size $k_s$ (Table 1) does not require a subscript since it is a fixed value across levels. Consider omitting the subscript.

Ok, done.

"As an illusion" in paragraph 215: misplaced use of the word "illusion". Consider replacing with "Illustration".

Done, sorry for the typo!

Table 2: Typo in "desactivated". Additionally, consider explaining the meaning of the rows of the table as some are not self evident, for example "refinement step".

The typo is fixed, and the following has been added to the manuscript.

In the case of a refinement step, the neural network is composed of two U-Nets: the first network provides an intermediate estimate of the missing data and the second U-Net uses the intermediate estimate and the original data to provide the final estimate. During training, the loss function is based on a weighted sum of the intermediate and final estimate. For inference, only the final estimate is used. The weights are considered as hyperparameters. More information is provided in Barth et al. (2022).

On "(corresponding to the probabilities between 3% and 97%)": Should this not be equal to "between 1.5% and 98.5%" since each interval has a weight of 1/65 implying, that the first rank covers realizations with a probability of occurrence between 0 and 0.015 and the last rank between 0.98 and 1? Therefore, the middle ranks exhibit a coverage between 0.015 and 0.98, correct?

Indeed, we changed the probabilities in the revised manuscript. (Originally we were considering two bins at the lower end and two bins at the high end to be affected, but we changed this assessment in the revised manuscript (in particular after adding the comparison with DINCAE)).

**References**

Barth, A., Alvera-Azcárate, A., Troupin, C., and Beckers, J.-M.: DINCAE 2.0: multivariate convolutional neural network with error estimates to reconstruct sea surface temperature satellite and altimetry observations, Geoscientific Model Development, https://doi.org/10.5194/gmd-2021-353, 2022.

Buizza, R., Leutbecher, M., and Isaksen, L.: Potential use of an ensemble of analyses in the ECMWF Ensemble Prediction System, Quarterly Journal of the Royal Meteorological Society, 134, 2051–2066, https://doi.org/https://doi.org/10.1002/qj.346, 2008.

Ohishi, S., Miyoshi, T., and Kachi, M.: An ensemble Kalman filter-based ocean data assimilation system improved by adaptive observation error inflation (AOEI), Geoscientific Model Development, 15, 9057–9073, https://doi.org/10.5194/gmd-15-9057-2022, 2022.

Simon, E. and Bertino, L.: Application of the Gaussian anamorphosis to assimilation in a 3-D coupled physical-ecosystem model of the North Atlantic with the EnKF: a twin experiment, Ocean Science, 5, 495–510, https://doi.org/10.5194/os-5-495-2009, 2009.

---

## Author Comment (AC2)

**1 Reviewer 2**

**1.1 General Comments**

This research work proposes a new method that can be applied in the reconstruction of missing data in geophysical datasets, and more specifically cloudy satellite images. The method is applied successfully to chlorophyll images, improving the skill of state of the art reconstruction methods. The paper is well written, scientifically consistent and contains enough novel contributions. Without doubt it deserves publication, but after some technical clarifications/corrections and a couple of small extensions of the results. Thinking in a broad audience dealing with the problem of missing data in geophysical datasets, some parts of the document might not be very reader friendly, and look biased towards the computer vision community. Considering the scope of this journal, an effort in that direction would be appreciated. The relatively small area considered is the main weakness of the study and introduces some concern about the applicability of the technique elsewhere. However, as the diffusion model is already trained using the complete Black Sea, it would be possible to extend the analysis by repeating the analysis (with the same hyper-parameters) for other areas (dynamically similar and not) to produce a figure like Figure 9 for multiple locations (see details later).

Identification of specific comments and technical corrections: P==page; L==line

We are grateful for the careful reading and the detailed comments from Reviewer 2. We have addressed the comments from the reviewer in the revised manuscript. We agree that the small area is a weakness of our study. In the revised manuscript, we applied this technique to 9 additional domains of the Black Sea and compared the results of the diffusion model with the result of DINCAE. For each of these domains we also computed the variogram (Figure 9 of the original manuscript) as the reviewer suggested. We found that, overall, the conclusion is robust when applying this technique to other areas of the Black Sea. More information is in the detailed reply below.

**1.2 Specific Comments**

P2L41: DINCAE refers to version in Barth et al., 2020; or alternatively to modified version in Barth et al., 2022? To both?

In fact, this statement refers to both versions. We added the reference to the 2020 and 2022 papers to clarify this point. Thank you for pointing this out.

P3L67-73: for the broad audience dealing with geophysical missing data reconstruction techniques, this could be made a bit less technical and more descriptive

We have expanded this section in the hope that this description is now more easily understood by the typical geophysical audience of this journal. The revised paragraph now reads:

In the classifier-free guidance algorithm (Ho and Salimans, 2022), the neural network denoising the images also depends explicitly on the class label. While training the neural network, this class label is sometimes replaced by a null label (i.e. a vector with all elements equal to zero). As a result the trained neural network can either denoise *any* image of the training dataset (when given the null label) or a specific subset of the training dataset (matching the provided label). During sampling the reverse diffusion is steered by a scaled difference between the noise predicted knowing the label and the noise predicted with a null label and therefore enhancing the similarity of the generated image with the provided label.

P3L86-P4L90: it is not clear if the mean and std are these of a pixel over the group of images, or alternatively are calculated image by image

We have normalized the images by using the mean and standard deviation of the entire training dataset. We clarified this in the revised manuscript.

P4L91-93: alpha and beta parameters are identified as diffusion step t dependent. True? Make that more explicit if it is the case. For the ease of understanding also make explicit that degradation level increases with increasing t

Yes, $\alpha$ and $\beta$ depend on the diffusion step. We modified the revised manuscript at different places to make it more explicit.

P4L106: implications for this application of "small steps sizes beta" not very clear

We added that this has to be understood in the limit where the discrete diffusion process tends to the continuous diffusion.

P6L135: image frequency is? 3h, 6h, daily...?

We indeed forgot to mention that this is a daily dataset. This is corrected in the revised manuscript.

P6L140: why 20%? Conservative approach? Sensitivity detected for certain threshold?

This is our first implementation of a denoising diffusion model so we adopted a conservative approach here. To our knowledge it is also the first implementation of a denoising diffusion model trained on incomplete satellite data. Standard diffusion models thus require 100 % of valid data. One should also consider that during training we mask additional points using the cloud mask of other images and that the loss function considers only these additional masked pixels (where the ground truth) is available. If we would use a lower threshold, say 10%, and apply an additional mask (with potentially up to 90% of missing data), there would be several instances of image without any pixels that can be considered in the loss function. This could significantly increase the training and with unclear benefits. Related to this question, is the remark from the other reviewer how does the network behave if there is a significant amount of missing data. We have grouped the results into ranges of cloud fraction and the highest is 85%-95%.

P6L138-143: training and validation datasets are defined here, but later also test dataset is referred.
and
P15L248-250: previous comments on the distinction of training, verification and test periods apply here in comparison with development phase mentioned now

It was actually defined on line 141 in the original manuscript:

The **validation** and **test** data range from September 1, 2021 to August 31, 2022 and from September 1, 2022 to August 31, 2023, respectively. [emphasis added]

In the revised manuscript, we splitted the sentence in two for clarity:

The validation dataset is composed of the 12 months of data between September 1, 2021 to August 31, 2022. The following 12 months (from September 1, 2022 to August 31, 2023) are used as test data.

> P6L140: is that the number of training images or the number after breaking the original figures in tiles?

Yes, we added this is the revised manuscript:

Only tiles with at least 20% valid data (i.e. non-clouded pixels) are used for training to reduce training time. In total, there are 851926 images (after splitting the data into tiles) for training.

> P6L139: reason for the 64x64 tile splitting is given later; here is confusing without justification; say choice is justified later?

We agree and changed the order in this section. In the revised manuscript we start the relevant paragraph with the justification.

> P6L146: what is meant with DINCAE being only applied with a fixed location? Does it mean that while the diffusion model is trained using data over the complete area DINCAE is applied only to the small box in Figure 2, and hence they are only compared over that small location?

Yes, this is correct. While we could have trained DINCAE also with data from other locations, DINCAE has only been tested and validated so far when trained with a fixed location. We preferred to keep the implementation of DINCAE as it was discussed in the previous published papers. The diffusion model could not be trained using a fixed location as discussed on lines 216 - 219 of the original manuscript.

> P6L146-147: that justifies the use of a small area for testing purposes, but why not extend the comparison to other locations in the area (by for example adding extra small areas of the same size, maybe also with some overlapping?)
> and
> P15L248: regarding proposal of extension to other areas made in the "general comments", this would imply training of DINCAE for such areas in this step.

In the revised manuscript we applied the diffusion and DINCAE to addition regions in the Black Sea. For overlapping domains, one can compute ensemble statistics such as mean and variances provided by the diffusion model which should be consistent from one domain to the next overlapping domain. However, this is not the case of the individual ensemble members which are currently independently. We plan to address this limitation in future follow-up work.

The following as been added to the manuscript:

Further domains are considered to test the applicability of the trained diffusion model in comparison with DINCAE to explore the different dynamical regimes. In Figure 13, the domain used previously is labeled as 1, and the additional domains are labeled 2 to 10. For each of these domains DINCAE is trained using only the data from the corresponding domain using the hyperparameters presented in Table B1. As the diffusion model is trained using 64 x 64 tiles from the whole Black Sea, it is not trained again but used only in the inference mode. The RMS error for each domain is shown in table 1 and the corresponding variogram can be seen in Figure 14. Overall the results from the previous test on the first domain are also applicable to other domains. The RMS error of the diffusion model is lower than the corresponding RMS error of DINCAE except for domain 7. At the same time, the variance for all domains across different scales is more realistic for the diffusion model.

[Figure]

Figure 13: Additional domains where the diffusion model is applied (domain 2 to domain 10)

[Figure]

Figure 14: Variogram for the independent test data

Table 1: RMS error relative to the independent test data for different domains.

| domain | RMS DINCAE | RMS Diffusion Model | std(obs) |
|---|---:|---:|---:|
| 1 | 0.175 | 0.163 | 0.331 |
| 2 | 0.159 | 0.058 | 0.226 |
| 3 | 0.225 | 0.056 | 0.211 |
| 4 | 0.162 | 0.155 | 0.253 |
| 5 | 0.162 | 0.074 | 0.251 |
| 6 | 0.182 | 0.143 | 0.353 |
| 7 | 0.090 | 0.096 | 0.295 |
| 8 | 0.119 | 0.062 | 0.286 |
| 9 | 0.189 | 0.149 | 0.442 |
| 10 | 0.116 | 0.111 | 0.244 |
| median | 0.158 | 0.107 | 0.289 |

> P6L146-149: why is this relevant? Why proceed this way? It seems relevant for the image reconstruction community, but is that the case for an audience dealing with geophysical data reconstruction?

This sentence has been removed in the revised manuscript.

> P9L165: eq. 1 also?

Yes, equation 1 should also be mentioned here. The manuscript is updated. Thank you

> P9L165: how does it ensure spatial coherence?

To clarify this point we added the following to the paragraph in question:

It is important to note that all operations in the training and sampling algorithms (equations 1, 10 and 11) are only pointwise operations (i.e., operations that apply to each grid cell independently) that do not involve the neighboring grid cells, except for the neural network which ensures spatial coherence. The spatial coherence is mainly due to the convolutional layers whose weights have been trained to provide the same spatial structure as in the training dataset.

> P9L173-174: it refers to the pixels of the added cloud mask? T is randomly selected for each pixel in that mask or the whole mask? From figure 4 and explanation in P11L196-197 it looks that is shared for all pixels in the mask...

Yes, this is correct. The step T is shared among all additional masked pixels. This has been clarified in the text:

The stage of degradation $t$ of these pixels is randomly chosen between 1 and $T$ but applied uniformly to all withheld pixels. This is important because the noise is reduced progressively during inference and the neural network needs to know how to handle intermediate degradation levels.

> P9L177 and Figure 4 caption: "Scaled diffusion step" of figure lacks explanation at this point (comes later); a basic description in the figure caption would help

We agree and extended the caption with the information: "The diffusion step $t$ ($0 \leq t \leq T$) is scaled linearly to the interval $-\frac{1}{2}$ and $\frac{1}{2}$"

> P10L180-182 and Figure 4: the way "predicted noise" in figure 4 is created from the "partially corrupted image", based on the description within this line, is not easy to understand and could be more explicit (how the neural network operates to create the figure)

The output of the neural network is a 2D field aiming to predict the noise that has been added to the input. The neural network can predict the added noise because it learned the typical spatial structures in the training dataset and it is able to recognise them even in a corrupted image. At a first approximation, the neural network acts like a high-pass filter to identify the noise, which is then removed iteratively during sampling.

> P10 Figure 5: this figure is not mentioned in the text and it is not clear how it interacts with the neural network; if it is part of it or a separated process...

A reference to this figure has been added by addressing the following issue here below.

> P11L198 would indicate that the training operates in the forward direction, while P5L127-129 that it is the same trained network that is applied, but in the reverse mode, to produce a ensemble of possible reconstructed versions... True? Make this clearer

During training, noise is intentionally added to the image (advancing from diffusion step $l$ to $l+1$) and the neural network is trained to predict the noise, allowing it to denoise the image and go from step $l+1$ back to $l$.

The following has been changed in the sampling section of the revised manuscript:

After training the neural network, the missing data in the validation and test dataset are reconstructed. Every clear pixel of the input image is considered to be in the non-degraded state $t=0$ and all other pixels (clouded or on land) are in the fully degraded state $t=T$ and initialized with normal distributed random values. For the later pixels, the reverse diffusion process is used iteratively (going from step $l+1$ to $l$) to reduce their noise keeping the originally present pixels unchanged (Figure 5).

> P11L197: then -1/2 refers to initial or non degraded while 1/2 is fully degraded? Make that explicit

This is correct. We added this information in the revised manuscript.

> P11L205-212: No comment about future or present public code availability?

The source code for training and inference of the neural network is now available at the address: https://github.com/gher-uliege/DINDiff.jl.

> P11L211-212: training and validation datasets were presented in "Data" section; does this refer to "validation" dataset? Is there a third "test" dataset?

As mentioned in another comment above, the test data were actually defined on line 141 (data section) in the original manuscript. In the revised manuscript, we introduced the test and validation data in two separate sentences for clarity.

P12L222: same as previous comments

As mentioned before, the paragraph has been revised:

After training the neural network, the missing data in the validation and test dataset are reconstructed. Every clear pixel of the input image is considered to be in the non-degraded state $t = 0$ and all other pixels (clouded or on land) are in the fully degraded state $t = T$ and initialized with normal distributed random values. For the later pixels, the reverse diffusion process is used iteratively (going from step $l + 1$ to $l$) to reduce their noise keeping the originally present pixels unchanged (Figure 5).

We hope that this is now clearer.

P12L225: comment about "P9L165" on the spatial coherence made before clarifies here, maybe is enough to mention that only here

We added specifically that the convolutional layers of the U-Net ensure spatial coherence in the method section.

P13L229: any reason to select that particular ensemble size?

We chose 64 ensemble members as a compromise between the diversity of ensemble members and computational time and guided by the fact that the ECMWF real-time S2S forecasts use 51 members for its ensemble forecast. We updated the manuscript accordingly.

P13L241: Blue for zero std? Mask out using another out of the color bar (white,...)?

A standard deviation of zero for initially present pixels is indeed intended. All ensemble members will be identical where the pixels are initially present. As a result the corresponding standard deviation is zero. The ensemble members are only different where the initial pixel is clouded. The area in uniform blue is thus not a mask and indeed a true zero.

We expanded the paragraph in the following way in the revised manuscript:

For every ensemble member, the interpolated fields in the pixels for which we have valid values in the input data is, per construction, identical to the initial input value. The ensemble standard deviation at these locations is thus consequently equal to zero.

P15L257-258: RMS error of the diffusion model is calculated as the ensemble average of individual RMS values?

In fact, the RMS error of the diffusion model is based on the ensemble mean. We added this information to the revised manuscript.

P15L259-260 and Tables 3 and 4: It would be interesting to add some metric about the intra-ensemble variability of the RMS for the diffusion model (in Tables 3 and 4): RMS_max and RMS_min and/or variance/std of the RMS inside the ensemble. This would provide a basic idea about the quality of individual reconstructions inside the ensemble.

As suggested, we compute the RMS error for every ensemble member individually and compute the minimum and maximum of these error statistics. It is important to note that the averaging operation for the RMS runs over all space and time dimensions (but obviously considering only pixels where there are actual measurements but whose value has been withheld). The revised paragraph now reads:

In all cases, the bias is relatively low and does not contribute significantly to the RMS error. The RMS error of the diffusion model (based on the ensemble mean) is slightly smaller than the RMS error of DINCAE for development and test datasets. However, as expected the RMS error of every ensemble member individually is substantially larger than the RMS error of the ensemble mean. Given that the RMS error is computed over all time instances, the RMS error for a single ensemble member is relatively stable. The maximum and minimum RMS error among the 64 ensemble members are 0.202 and 0.211 $\log_{10}$ mg m$^{-3}$ respectively.

P18L270: "randomly chosen locations" including a certain number of comparisons? all possible?

We agree that this was not clear enough. We added the following part:

Here we are considering a variogram only as a function of distance $h = \|\mathbf{x}_1 - \mathbf{x}_2\|$, which allows us to estimate the variogram numerically by computing the squared differences for the field at randomly chosen locations. These squared differences are averaged over bins of distances using all time instances of the validation and test datasets. As many different random locations were chosen until there are at least 10000 pairs for each bin of distance.

P18L271-272: that means that comparisons are made for individual members and then averaged over the ensemble to produce the metric value?

Yes, this is correct. Following has been added to the revised manuscript to clarify this:

For the diffusion model, the variogram is deduced using the individual ensemble members, and the averaging in equation (12) is done also over different ensemble members.

P18L278: "does not converge to zero" for the original data and the diffusion model, not for DINCAE?

This is correct. We added the following to the text:

DINCAE effectively removes (or significantly reduces) the spatially uncorrelated white noise and therefore the corresponding variogram shows a clear tendency towards zero for smaller distances.

P20L295-297: meaning of the over estimated upper and lower limits?

The corresponding paragraph has been rewritten and now reads:

... It can be seen that the error statistics of the diffusion model are closer to the ideal flat curve for the diffusion model than for DINCAE. This shows that the probabilities produced by the diffusion model are marginally reliable, except for the tails of the marginal PDF (first and last bin, corresponding to the probabilities between 1.5% and 98.5%) where the produced ensemble is underdispersive.

P20-21L299-320: Although interesting, the analysis of the CRPS seem somehow weak without any reference to compare with...

The other reviewer made a similar comment and asked to produce the relevant scores for DINCAE. While DINCAE is not able to produce a full ensemble (or full pdf), the CRPS score relies only on the marginal pdf, which can be provided by DINCAE. The corresponding Talagrand diagrams have also been produced. The table with the CRPS results and the accompanying discussion has been extended in the revised manuscript.

P22L336-342: multivariate reconstructions are mentioned, but what about univariate reconstructions like for Sea Surface Temperature or Sea Surface Salinity, for instance, how is the method expected to behave?

We are optimistic that the method can also be applied to sea surface temperature, as the spatial and temporal scales in these images are more coherent and less patchy than those in chlorophyll data, which should help in the reconstruction. However, for sea surface salinity having large area (almost) constantly masked due to radio frequency interference could pose a problem for the diffusion model as it is not clear if the neural network can learn the appropriate scales for these regions.

**1.3 Technical corrections**

P2L33-35: the reference to the error of initially missing values is confusing

This sentence has been rephrased as:

For satellite images where all missing data have been reconstructed, it is clear that the error of the reconstructed and initial missing pixels is typically larger than the error of the original pixels.

P2L51-53: hard to understand, could be clearer? An example maybe?

We have added the following example in the hope that this is now clearer:

Since multiple images would be consistent with the partial information present, a neural network trained to minimize e.g. the mean square error, would then implicitly produce the average of all these possible states. For example, if the exact position of a front is not visible in a satellite image, a reconstructed image would have the tendency to smooth out the front as it is implicitly the average of multiple images with the front in different positions. Consequently, this means that small scale information cannot be adequately retained.

**References**

Ho, J. and Salimans, T.: Classifier-Free Diffusion Guidance, https://doi.org/10.48550/arXiv.2207.12598, 2022.